# GRAM-GAUSS-NEWTON METHOD: LEARNING OVER-PARAMETERIZED NEURAL NETWORKS FOR REGRESSION PROBLEMS

## ABSTRACT

First-order methods such as stochastic gradient descent (SGD) are currently the standard algorithm for training deep neural networks. Second-order methods, despite their better convergence rate, are rarely used in practice due to the prohibitive computational cost in calculating the second-order information. In this paper, we propose a novel Gram-Gauss-Newton (GGN) algorithm to train deep neural networks for regression problems with square loss. Our method draws inspiration from the connection between neural network optimization and kernel regression of neural tangent kernel (NTK). Different from typical second-order methods that have heavy computational cost in each iteration, GGN only has minor overhead compared to first-order methods such as SGD. We also give theoretical results to show that for sufficiently wide neural networks, the convergence rate of GGN is *quadratic*. Furthermore, we provide convergence guarantee for mini-batch GGN algorithm, which is, to our knowledge, the first convergence result for the mini-batch version of a second-order method on overparameterized neural networks. Preliminary experiments on regression tasks demonstrate that for training standard networks, our GGN algorithm converges much faster and achieves better performance than SGD.

## 1 INTRODUCTION

First-order methods such as Stochastic Gradient Descent (SGD) are currently the standard choice for training deep neural networks. The merit of first-order methods is obvious: they only calculate the gradient and therefore are computationally efficient. In addition to better computational efficiency, SGD has even more advantages among the first-order methods. At each iteration, SGD computes the gradient only on a mini-batch instead of all training data. Such randomness introduced by sampling the mini-batch can lead to better generalization (Hardt et al., 2015; Keskar et al., 2016; Masters & Luschi, 2018; Mou et al., 2017; Zhu et al., 2018) and better convergence (Ge et al., 2015; Jin et al., 2017a;b), which is crucial when the function class is highly overparameterized deep neural networks. Recently there is a huge body of works trying to develop more efficient first-order methods beyond SGD (Duchi et al., 2011; Kingma & Ba, 2014; Luo et al., 2019; Liu et al., 2019).

Second-order methods, despite their better convergence rate, are rarely used to train deep neural networks. At each iteration, the algorithm has to compute second order information, for example, the Hessian or its approximation, which is typically an $m$ by $m$ matrix where $m$ is the number of parameters of the neural network. Moreover, the algorithm needs to compute the inverse of this matrix. The computational cost is prohibitive and usually it is not even possible to store such a matrix.

Recently, there is a series of works (Du et al., 2018b;a; Zou et al., 2018; Allen-Zhu et al., 2018a; Oymak & Soltanolkotabi, 2018; Arora et al., 2019b;a; Cao & Gu, 2019; Zou & Gu, 2019) considering the optimization of neural networks based on the idea of neural tangent kernel (NTK) (Jacot et al., 2018). Roughly speaking, the idea of NTK is to linearly approximate the output of a network w.r.t. the parameters at a local region, thus resulting in a kernel feature map, which is the gradient of the output w.r.t. the parameters. Jacot et al. (2018) shows that when the width of the network tends to infinity, the NTK tends to be unchanged during gradient flow since the network only needs $o(1)$ change (goes to zero as the width tends to infinity) of parameter to fit the data, so the change of kernel

is also $o(1)$. As for finite-width networks, however, the NTK changes and previous works (Allen-Zhu et al., 2018b; Arora et al., 2019a) show that for sufficiently wide neural networks the optimization dynamic of GD/SGD is equivalent to that of using GD/SGD to solve an NTK kernel regression problem where the kernel is slowly evolving (see Lemma 1 in Section 2 for a precise description). A natural question then arises:

*Can we gain acceleration by directly solving kernel regression w.r.t. the NTK at each step?*

In this paper, we give a positive answer to this question and reveal the connection between NTK regression and Gauss-Newton method. We propose a novel optimization method – the Gram-Gauss-Newton (GGN) method. Instead of doing gradient descent, GGN solves the kernel regression w.r.t. the NTK at each step of the optimization. Following this idea, we theoretically prove that for overparameterized networks, GGN enjoys quadratic convergence compared to the linear rate of gradient descent.

Besides theoretical fast convergence rate, GGN is also very efficient in practice. In fact, GGN is implicitly a reformulation of the Gauss-Newton method (see Section 3.1 for details) which is a classic second-order algorithm often used for solving nonlinear regression problems with square loss. In the Gauss-Newton method, one uses $\mathbf{J}^\top \mathbf{J}$ as an approximation of the Hessian (see Section 2 for a formal description) where $\mathbf{J}$ is the Jacobian matrix. However, the original Gauss-Newton method faces challenges when used for training deep neural networks. Most seriously, the size of the approximate Hessian $\mathbf{J}^\top \mathbf{J}$ is $m$ by $m$, where $m$ is the number of parameters of the neural network. Moreover, for overparameterized neural networks, $\mathbf{J}^\top \mathbf{J}$ is not invertible, which may make the algorithm intractable for training commonly-used neural networks.

GGN bypasses the difficulty stated above as follows. Instead of using $\mathbf{J}^\top \mathbf{J}$ as approximate Hessian and applying Newton-type method, each step of GGN only involves the Gram matrix $\mathbf{J}\mathbf{J}^\top$ whose size is $n$ by $n$ where $n$ is the number of data. Furthermore, as already mentioned, to get better generalization performance, it is crucial to use mini-batch to introduce sampling noise when calculating derivatives. Therefore, like SGD, we also use mini-batch in GGN. In this case, the size of the Gram matrix further reduces to $b$ by $b$, where $b$ is the batch size. Though conventional wisdom may suggest that applying mini-batch scheme to second-order methods will introduce biased estimation of the accelerated gradient direction, we give the first convergence result for mini-batch second-order method on overparameterized networks. Regarding computational complexity, we show that at each iteration, the overhead of GGN is small compared to SGD: the extra computation of GGN is mainly the matrix product $\mathbf{J}\mathbf{J}^\top$ and the inverse of this matrix whose size is small for a mini-batch. Detailed analyses can be found in Section 3.3. We next conduct experiments on two regression tasks to study the effectiveness of the GGN algorithm. We demonstrate that in these two real applications, using a practical neural network (e.g., ResNet-32) with standard width, our proposed GGN algorithm can converge faster and achieve better performance than several baseline algorithms.

## 1.1 RELATED WORKS

Despite the prevalence of first-order methods for training deep neural networks, there have been continuing efforts in developing practical second-order methods (Becker et al., 1988; Pascanu & Bengio, 2013). We summarize some of these works below.

The main approach for these methods is to develop delicate approximations of the second-order information matrix so that the update direction can be computed as efficiently as possible. For example, Botev et al. (2017) proposed a recursive block-diagonal approximation of the Hessian. The blocks are Kronecker factored and can be efficiently computed and inverted. Grosse and Martens in a series of works developed the K-FAC method (Martens & Grosse, 2015; Grosse & Martens, 2016). The key idea is a Kronecker-factored approximation of the Fisher information matrix, which is used as the second-order matrix in natural gradient methods. These works received considerable attention and have been further improved (Wu et al., 2017; George et al., 2018; Martens et al., 2018). Bernacchia et al. (2018) derived an exact expression of the natural gradient update, but only works for linear networks. Different from all these works, our GGN algorithm does not try to approximate the second-order matrix whose size is inevitably huge. Instead, we present an easy-to-compute solution of the updating direction, reducing the computational cost significantly. One exceptional concurrent work Ren & Goldfarb (2019) also aims to use the exact Gauss-Newton update. They focus on reducing the complexity of inverting approximate Hessian by Sherman-Morrison-Woodbury

Formula and require subtle implementation tricks to use backpropagation. In contrast, GGN has simpler update rule and better guarantee for neural networks.

In a concurrent and independent work, Zhang et al. (2019a) showed that natural gradient method and K-FAC have a linear convergence rate for sufficiently wide networks in full-batch setting. In contrast, our method enjoys a higher-order (quadratic) convergence rate guarantee for overparameterized networks, and we focus on developing a practical and theoretically sound optimization method. We also reveal the relation between our method and NTK kernel regression, so using results based on NTK (Arora et al., 2019b), one can easily give generalization guarantee of our method. Another independent work (Achiam et al., 2019) proposed a preconditioned Q-learning algorithm which has similar form of our update rule. Unlike the methods considered in Zhang et al. (2019a); Achiam et al. (2019) which contain the learning rate that needed to be tuned, our derivation of GGN does not introduce a learning rate term (or understood as suggesting that the learning rate can be fixed to be 1 to get good performance which is verified in Figure 2 (c)).

## 2    NEURAL TANGENT KERNEL AND THE CLASSIC GAUSS-NEWTON METHOD FOR NONLINEAR LEAST SQUARES REGRESSION

Nonlinear least squares regression problem is a general machine learning problem. Given data pairs $\{\mathbf{x}_i, y_i\}_{i=1}^n$ and a class of nonlinear functions $f$, e.g. neural networks, parameterized by $\mathbf{w}$, the nonlinear least squares regression aims to solve the optimization problem

$$\min_{\mathbf{w} \in \mathbb{R}^m} L(\mathbf{w}) = \frac{1}{2} \sum_{i=1}^n (f(\mathbf{w}, \mathbf{x}_i) - y_i)^2. \tag{1}$$

In the seminal work (Jacot et al., 2018), the authors consider the case when $f$ is a neural network with infinite width. They showed that optimization on this problem using gradient flow involves a special kernel which is called neural tangent kernel (NTK). The follow-up works further extended the relation between optimization and NTK which can be concluded in the following lemma:

**Lemma 1** (Lemma 3.1 in Arora et al. (2019a), see also Dou & Liang (2019); Mei et al. (2019)).
*Consider optimizing problem* (1) *by gradient descent with infinitesimally small learning rate:* $\frac{d\mathbf{w}_t}{dt} = -\nabla L(\mathbf{w}_t)$. *where* $\mathbf{w}_t$ *is the parameters at time* $t$. *Let* $\mathbf{f}_t = (f(\mathbf{w}_t, \mathbf{x}_i))_{i=1}^n \in \mathbb{R}^n$ *be the network outputs on all* $\mathbf{x}_i$*'s at time* $t$, *and* $\mathbf{y} = (y_i)_{i=1}^n$ *be the desired outputs. Then* $\mathbf{f}_t$ *follows the following evolution:*

$$\frac{d\mathbf{f}_t}{dt} = -\mathbf{G}_t \cdot (\mathbf{f}_t - \mathbf{y}), \tag{2}$$

*where* $\mathbf{G}_t$ *is an* $n \times n$ *positive semidefinite matrix, i.e.* **the Gram matrix w.r.t. the NTK at time** $t$, *whose* $(i, j)$*-th entry is* $\langle \nabla_\mathbf{w} f(\mathbf{w}_t, \mathbf{x}_i), \nabla_\mathbf{w} f(\mathbf{w}_t, \mathbf{x}_j) \rangle$.

The key idea of Jacot et al. (2018) and its extensions (Du et al., 2018b;a; Zou et al., 2018; Allen-Zhu et al., 2018a; Oymak & Soltanolkotabi, 2018; Lee et al., 2019; Yang, 2019; Arora et al., 2019b;a; Cao & Gu, 2019; Zou & Gu, 2019) is that when the network is sufficiently wide, the Gram matrix at initialization $\mathbf{G}_0$ is close to a fixed positive definite matrix defined by the infinite-width kernel and $\mathbf{G}_t$ *is close to* $\mathbf{G}_0$ *during training for all* $t$. Under this situation, $\mathbf{G}_t$ remains invertible, and the above dynamics is then identical to the dynamics of solving *kernel regression* with gradient flow w.r.t. the current kernel at time $t$. In fact, Arora et al. (2019a) rigorously proves that a fully-trained sufficiently wide ReLU neural network is equivalent to the kernel regression predictor.

As pointed out in Chizat & Bach (2018), the idea of NTK can be summarized as a linear approximation using first order Taylor expansion. We give an example of this idea on the NTK at initialization:

$$f(\mathbf{w}, \mathbf{x}_i) - f(\mathbf{w}_0, \mathbf{x}_i) \approx \nabla_\mathbf{w} f(\mathbf{w}_0, \mathbf{x}_i) \cdot (\mathbf{w} - \mathbf{w}_0), \tag{3}$$

where $\nabla_\mathbf{w} f(\mathbf{w}_0, \mathbf{x})$ can then be viewed as an explicit expression of feature map at $\mathbf{x}$, $\mathbf{w} - \mathbf{w}_0$ is the parameter in reproducing kernel Hilbert space (RKHS) induced by NTK and $f(\mathbf{w}, \mathbf{x}_i) - f(\mathbf{w}_0, \mathbf{x}_i)$ the target value.

The idea of linear approximation is also used in the classic Gauss-Newton method (Golub, 1965) to obtain an acceleration algorithm for solving nonlinear least squares problem (1). Concretely, at iteration $t$, Gauss-Newton method takes the following first-order approximation:

$$f(\mathbf{w}, \mathbf{x}_i) - f(\mathbf{w}_t, \mathbf{x}_i) \approx \nabla_\mathbf{w} f(\mathbf{w}_t, \mathbf{x}_i) \cdot (\mathbf{w} - \mathbf{w}_t), \tag{4}$$

where $\mathbf{w}_t$ stands for the parameter at iteration $t$. We note that this is also *the linear expansion for deriving NTK at time $t$*. According to Eq. (1) and (4), to update the parameter, one can instead solve the following problem.

$$\mathbf{w}_{t+1} = \operatorname*{argmin}_{\mathbf{w}} \frac{1}{2} \|\mathbf{f}_t + \mathbf{J}_t(\mathbf{w} - \mathbf{w}_t) - \mathbf{y}\|_2^2, \tag{5}$$

where $\mathbf{f}_t, \mathbf{y}$ have the same meaning as in Lemma 1, and $\mathbf{J}_t = (\nabla_{\mathbf{w}} f(\mathbf{w}_t, \mathbf{x}_1), \cdots, \nabla_{\mathbf{w}} f(\mathbf{w}_t, \mathbf{x}_n))^\top \in \mathbb{R}^{n \times m}$ is the Jacobian matrix.

A necessary and sufficient condition for $\mathbf{w}$ to be the solution of Eq. (5) is

$$(\mathbf{J}_t^\top \mathbf{J}_t) \cdot (\mathbf{w} - \mathbf{w}_t) = -\mathbf{J}_t^\top (\mathbf{f}_t - \mathbf{y}). \tag{6}$$

Below we will denote $\mathbf{H}_t := \mathbf{J}_t^\top \mathbf{J}_t \in \mathbb{R}^{m \times m}$. For under-parameterized model (i.e., the number of parameters $m$ is less than the number of data $n$), $\mathbf{H}_t$ is invertible, and the update rule is

$$\mathbf{w}_{t+1} = \mathbf{w}_t - \mathbf{H}_t^{-1} \mathbf{J}_t^\top (\mathbf{f}_t - \mathbf{y}). \tag{7}$$

This can also be viewed as an approximate Newton's method using $\mathbf{H}_t = \mathbf{J}_t^\top \mathbf{J}_t$ to approximate the Hessian matrix. In fact, the exact Hessian matrix is

$$\nabla_{\mathbf{w}}^2 \frac{1}{2} \sum_{i=1}^{n} (f(\mathbf{w}_t, \mathbf{x}_i) - y_i)^2 = \mathbf{J}_t^\top \mathbf{J}_t + \sum_{i=1}^{n} (f(\mathbf{w}_t, \mathbf{x}_i) - y_i) \nabla_{\mathbf{w}}^2 f(\mathbf{w}_t, \mathbf{x}_i). \tag{8}$$

In the case when $f$ is only mildly nonlinear w.r.t. $\mathbf{w}$ at data point $\mathbf{x}_i$'s, $\nabla_{\mathbf{w}}^2 f(\mathbf{w}_t, \mathbf{x}_i) \approx 0$, and $\mathbf{H}_t$ is close to the real Hessian. In this situation, the behavior of the Gauss-Newton method is similar to that of Newton's method, and thus can achieve a superlinear convergence rate (Golub, 1965).

## 3 THE GRAM-GAUSS-NEWTON METHOD

The classic second-order methods using approximate Hessian such as Gauss-Newton method described in the previous section face obvious difficulties dealing with the intractable approximate Hessian matrix when the regression model is an overparameterized neural network.

In Section 3.1, we develop a Gram-Gauss-Newton (GGN) method which is inspired by NTK kernel regression and does not require the computation of the approximate Hessian. In Section 3.2, we show that for sufficiently wide neural networks, GGN has quadratic convergence rate. In Section 3.3, we show that the additional computational cost (per iteration) of GGN compared to SGD is small.

### 3.1 THE GRAM-GAUSS-NEWTON METHOD FOR OVERPARAMETERIZED NEURAL NETWORKS

We now describe our GGN method to learn overparameterized neural networks for regression problems. As mentioned in the previous sections, for sufficiently wide networks, using gradient descent for solving the regression problem (1) has similar dynamics as using gradient descent for solving NTK kernel regression (Lemma 1) w.r.t. NTK at each step. However, one can also solve the kernel regression problem w.r.t. the NTK at each step immediately using the explicit formula of kernel regression. By explicitly solving instead of using gradient descent to solve NTK kernel regression, one can expect the optimization to get accelerated. We propose our Gram-Gauss-Newton (GGN) method to directly solve the NTK kernel regression with Gram matrix $\mathbf{G}_t$ at each time step $t$. Note that the feature map of NTK at time $t$, based on the approximation in Eq. (4), can be expressed as $x \mapsto \nabla_{\mathbf{w}} f(\mathbf{w}_t, \mathbf{x})$, and the linear parameters in RKHS are $\mathbf{w} - \mathbf{w}_t$, also the target is $f(\mathbf{w}, \mathbf{x}_i) - f(\mathbf{w}_t, \mathbf{x}_i)$. Therefore, the kernel (ridgeless) regression solution (Mohri et al., 2018; Liang & Rakhlin, 2018) of $\mathbf{J}_{t,S}(\mathbf{w} - \mathbf{w}_t) = (f(\mathbf{w}, \mathbf{x}_i) - f(\mathbf{w}_t, \mathbf{x}_i))$ w.r.t. $(\mathbf{w} - \mathbf{w}_t)$ gives the update

$$\mathbf{w}_{t+1} = \mathbf{w}_t - \mathbf{J}_{t,S}^\top \mathbf{G}_{t,S}^{-1} (\mathbf{f}_{t,S} - \mathbf{y}_S), \tag{9}$$

where $\mathbf{J}_{t,S}$ is the matrix of features at iteration $t$ computed on the training data set $S$ which is equal to the Jacobian , $\mathbf{f}_{t,S}$ and $\mathbf{y}_S$ are the vectorized outputs of neural network and the corresponding targets on $S$ respectively, and

$$\mathbf{G}_{t,S} = \mathbf{J}_{t,S} \mathbf{J}_{t,S}^\top$$

is the Gram matrix of the NTK on $S$.

One may wonder what is the relation between our derivation from NTK kernel regression and the Gauss-Newton method. We point out that for overparameterized models, there are infinitely many solutions of Eq. (5) but our update rule (9) essentially uses the *minimum norm* solution. In other words, the GGN update rule re-derives the Gauss-Newton method with the minimum norm solution. This somewhat surprising connection is due to the fact that in kernel learning, people usually choose a kernel with powerful expressivity, i.e. the dimension of feature space is large or even infinite. However, by the representer theorem (Mohri et al., 2018), the solution of kernel (ridgeless) regression lies in the $n$-dimensional subspace of RKHS and minimizes the RKHS norm. We refer the readers to Chapter 11 of Mohri et al. (2018) for details.

As mentioned in Section 1, the design of learning algorithms should consider not only optimization but also generalization. It has been shown that using mini-batch instead of full batch to compute derivatives is crucial for the learned model to have good generalization ability (Hardt et al., 2015; Keskar et al., 2016; Masters & Luschi, 2018; Mou et al., 2017; Zhu et al., 2018). Therefore, we propose a mini-batch version of GGN. The update rule is the following:

$$\mathbf{w}_{t+1} = \mathbf{w}_t - \mathbf{J}_{t,B_t}^\top \mathbf{G}_{t,B_t}^{-1}(\mathbf{f}_{t,B_t} - \mathbf{y}_{B_t}), \tag{10}$$

where $B_t$ is the mini-batch used at iteration $t$, $\mathbf{J}_{t,B_t}$ and $\mathbf{G}_{t,B_t}$ are the Jacobian and the Gram matrix computed using the data of $B_t$ respectively, and $\mathbf{f}_{t,B_t}, \mathbf{y}_{B_t}$ are the vectorized outputs and the corresponding targets on $B_t$ respectively. $\mathbf{G}_{t,B_t} = \mathbf{J}_{t,B_t}\mathbf{J}_{t,B_t}^\top$ is a very small matrix when using a typical batch size.

One difference between Eq. (10) and Eq. (7) is that our update rule only requires to compute the Gram matrix $\mathbf{G}_{t,B_t}$ and its inverse. Note that the size of $\mathbf{G}_{t,B_t}$ is equal to the size of the mini-batch and is typically very small. So this also greatly reduces the computational cost.

Using the idea of kernel ridge regression (which can also be viewed as Levenberg-Marquardt extension (Levenberg, 1944) of Gauss-Newton method), we introduce the following variant of GGN:

$$\mathbf{w}_{t+1} = \mathbf{w}_t - \mathbf{J}_{t,B_t}^\top(\lambda\mathbf{G}_{t,B_t} + \alpha\mathbf{I})^{-1}(\mathbf{f}_{t,B_t} - \mathbf{y}_{B_t}), \tag{11}$$

where $\lambda > 0$ is another hyper-parameter controlling the learning process. Our algorithm is formally described in Algorithm 1.

---

**Algorithm 1** (Mini-batch) Gram-Gauss-Newton Method

---

1: **Input**: Training dataset $S$. Hyper-parameters $\lambda$ and $\alpha$.
2: Initialize the network parameter $w_0$. Set $t = 0$.
3: **for** each iteration **do**
4:     Fetch a mini-batch $B_t$ from the dataset.
5:     Calculate the Jacobian matrix $\mathbf{J}_{t,B_t}$.
6:     Calculate the Gram matrix $\mathbf{G}_{t,B_t} = \mathbf{J}_{t,B_t}\mathbf{J}_{t,B_t}^\top$.
7:     Update the parameter by $w_{t+1} = w_t - \mathbf{J}_{t,B_t}^\top(\lambda\mathbf{G}_{t,B_t} + \alpha\mathbf{I})^{-1}(\mathbf{f}_{t,B_t} - \mathbf{y}_{B_t})$.
8:     $t = t + 1$.
9: **end for**

---

### 3.2 Convergence Analysis for Overparameterized Neural Networks

In this subsection, we show that for two-layer neural networks, if the width is sufficiently large, then: (1) Full-batch GGN converges with quadratic convergence rate. (2) Mini-batch GGN converges linearly. (For clarity, here we only present a proof for two-layer neural networks, but we believe that it is not hard for the conclusion to be extended to deep neural networks using the techniques developed in Du et al. (2018a); Zou & Gu (2019)).

As we explained through the lens of NTK, the result is a consequence of the fact that for wide enough neural networks, if the weights are initialized according to a suitable probability distribution, then with high probability the output of the network is close to a linear function w.r.t. the parameters (but nonlinear w.r.t. the input of the network) in a neighborhood containing the initialization point and a global optimum. Although the neural networks used in practice are far from that wide, this still motivates us to design the GGN algorithm.

**Neural network structure.** We use the following two-layer network

$$f(\mathbf{w}, \mathbf{x}) = \frac{1}{\sqrt{M}} \mathbf{a}^\top \sigma(\mathbf{W}^\top \mathbf{x}) = \frac{1}{\sqrt{M}} \sum_{r=1}^{M} \mathbf{a}_r \sigma(\mathbf{w}_r \mathbf{x}), \tag{12}$$

where $\mathbf{x} \in \mathbb{R}^d$ is the input, $M$ is the network width, $\mathbf{W} = (\mathbf{w}_1^\top, \cdots, \mathbf{w}_M^\top)^\top$ and $\sigma(\cdot)$ is the activation function. Each entry of $\mathbf{W}$ is i.i.d. initialized with the standard Gaussian distribution $\mathbf{w}_r \sim \mathcal{N}(0, \mathbf{I}_d)$ and each entry of $\mathbf{a}$ is initialized from the uniform distribution on $\{\pm 1\}$. Similar to Du et al. (2018b), we only train the network on parameter $\mathbf{W}$ just for the clarity of the proof. We also assume the activation function $\sigma(\cdot)$ is $\ell$-Lipschitz and $\beta$-smooth, and $\ell$ and $\beta$ are regarded as $O(1)$ absolute constants.

The key finding, as pointed out in Jacot et al. (2018); Du et al. (2018b;a), is that under such initialization, the Gram matrix $\mathbf{G}$ has an asymptotic limit, which is, under mild conditions (e.g. input data not being degenerate etc., see Lemma F.2 of Du et al. (2018a)), a positive definite matrix

$$\mathbf{K}(\mathbf{x}_i, \mathbf{x}_j) = \mathbb{E}_{\mathbf{w} \sim \mathcal{N}(0, \mathbf{I})} \left[ \mathbf{x}_i^\top \mathbf{x}_j \sigma'(\mathbf{w}\mathbf{x}_i) \sigma'(\mathbf{w}\mathbf{x}_j) \right]. \tag{13}$$

**Assumption 1** (Least Eigenvalue of the Limit of Gram Matrix). *We assume the matrix $\mathbf{K}$ defined in (13) above is positive definite, and denote its least eigenvalue as*

$$\lambda_0 = \lambda_{\min}(\mathbf{K}) > 0.$$

Now we are ready to state our theorem of full-batch GGN:

**Theorem 1** (Quadratic Convergence of Full-batch GGN on Overparameterized Neural Networks). *Assume Assumption 1 holds. Assume the scale of the data is $\|\mathbf{x}_i\|_2 = O(1)$, $|y_i| = O(1)$ for $i \in \{1, \cdots, n\}$. If the network width*

$$M = \Omega \left( \max \left( \frac{n^4}{\lambda_0^4}, \frac{n^2 d \log(16n/\delta)}{\lambda_0^2} \right) \right),$$

*then with probability $1 - \delta$ over the random initialization, the full-batch version of GGN whose update rule is given in Eq. (9) satisfies the following:*

*1) The Gram matrix $\mathbf{G}_{t,S}$ at each iteration is invertible;*

*2) The loss converges to zero in a way that*

$$\|\mathbf{f}_{t+1} - \mathbf{y}\|_2 \le \frac{C}{\sqrt{M}} \|\mathbf{f}_t - \mathbf{y}\|_2^2 \tag{14}$$

*for some $C$ that is independent of $M$, which is a second-order convergence.*

For the mini-batch version of GGN, by the analysis of its NTK limit, the algorithm is essentially doing serial subspace correction (Xu, 2001) on subspaces induced by mini-batch. So mini-batch GGN is similar to the *Gauss-Siedel method* (Golub & Van Loan, 1996) applied to solving systems of linear equations, as shown in the proof of the following theorem. Similar to the full batch situation, GGN takes the exact solution of the "kernel regression problem on the subspace" which is faster than just doing a gradient step to optimize on the subspace. Moreover, we note that existing results of the convergence of SGD on overparameterized networks usually use the idea that when the step size is bounded by a quantity related to smoothness, the SGD can be reduced to GD. However, our analysis takes a different way from the analysis of GD, thus does not rely on small step size.

In the following, we denote $\mathbf{G}_0 \in \mathbb{R}^{n \times n}$ as the initial Gram matrix. Let $n = bk$, where $b$ is the batch size and $k$ is the number of batches, and let

$$\mathbf{G}_{0,ij} := \mathbf{G}_0((i-1)b + 1 : ib, (j-1)b + 1 : jb)$$

be the $(i, j)$-th $b \times b$ block of $\mathbf{G}_0$. We define the iteration matrix

$$\mathbf{A} = \mathbf{L}^\top (\mathbf{D} - \mathbf{L})^{-1} \in \mathbb{R}^{n \times n}, \tag{15}$$

where

$$\mathbf{D} = \begin{bmatrix} \mathbf{G}_{0,11} & \mathbf{0} & \cdots & \mathbf{0} \\ \mathbf{0} & \mathbf{G}_{0,22} & \cdots & \mathbf{0} \\ \vdots & \vdots & \ddots & \vdots \\ \mathbf{0} & \mathbf{0} & \ddots & \mathbf{G}_{0,kk} \end{bmatrix}, \quad \mathbf{L} = - \begin{bmatrix} \mathbf{0} & \mathbf{0} & \cdots & \mathbf{0} \\ \mathbf{G}_{0,21} & \mathbf{0} & \cdots & \mathbf{0} \\ \vdots & \vdots & \ddots & \vdots \\ \mathbf{G}_{0,k1} & \mathbf{G}_{0,k2} & \ddots & \mathbf{0} \end{bmatrix},$$

represents the block-diagonal and block-lower-triangular parts of $\mathbf{G}_0$. We will show that the convergence of mini-batch GGN is highly related to the spectral radius of $\mathbf{A}$. To simplify the proof, we make the following mild assumption on $\mathbf{A}$:

**Assumption 2** (Assumption on the Iteration Matrix). *Assume the matrix $\mathbf{A}$ defined in (15) above is diagonalizable. So we choose an arbitary diagonalization of $\mathbf{A}$ as $\mathbf{A} = \mathbf{P}^{-1}\mathbf{Q}\mathbf{P}$ and denote*

$$\mu := \|\mathbf{P}\|_2 \left\|\mathbf{P}^{-1}\right\|_2.$$

We note that Assumption 2 is only for the sake of simplicity. Even if it does not hold, an infinitesimally small perturbation can make any matrix diagonalizable, and it will not affect the proof.

Now we are ready to state the theorem for mini-batch GGN.

**Theorem 2** (Convergence of Mini-batch GGN on Overparameterized Neural Networks). *Assume Assumption 1 and 2 hold. Assume the scale of the data is $\|\mathbf{x}_i\|_2 = O(1)$, $|y_i| = O(1)$ for $i \in \{1, \cdots, n\}$. We use the mini-batch version of GGN whose update rule is given in Eq.(10), and the batch $B_t$ is chosen sequentially and cyclically with a fixed batch size $b$ and $k = n/b$ updates per epoch. If the network width*

$$M = \max\left(\Omega\left(\frac{\mu^2 n^{18}}{\lambda_0^{16}}\right), \Omega\left(\frac{n^2 d \log(16n/\delta)}{\lambda_0^2}\right)\right),$$

*then with probability $1 - \delta$ over the random initialization, we have the following:*

*1) The Gram matrix $\mathbf{G}_{t,B_t}$ at each iteration is invertible;*

*2) The loss converges to zero in a way that after $T$ epochs, we have*

$$\|\mathbf{f}_{Tk} - \mathbf{y}\|_2 \leq \mu\sqrt{n}\left(1 - \Omega\left(\frac{\lambda_0^2}{n^2}\right)\right)^T. \tag{16}$$

**Proof sketch for Theorem 1 and 2.** Denote $\mathbf{J}_t = \mathbf{J}(\mathbf{W}_t)$ and

$$\mathbf{J}_{t,t+1} = \int_0^1 \mathbf{J}((1-s)\mathbf{W}_t + s\mathbf{W}_{t+1})ds.$$

For the full-batch version, we have

$$
\begin{aligned}
\|\mathbf{f}_{t+1} - \mathbf{y}\|_2 &= \|\mathbf{f}_t - \mathbf{y} + \mathbf{J}_{t,t+1}(\text{vec}(\mathbf{W}_{t+1}) - \text{vec}(\mathbf{W}_t))\|_2 \\
&= \left\|\mathbf{f}_t - \mathbf{y} - \mathbf{J}_{t,t+1}\mathbf{J}_t^\top \mathbf{G}_t^{-1}(\mathbf{f}_t - \mathbf{y})\right\|_2 \\
&= \left\|(\mathbf{J}_t - \mathbf{J}_{t,t+1})\mathbf{J}_t^\top \mathbf{G}_t^{-1}(\mathbf{f}_t - \mathbf{y})\right\|_2 \\
&\leq \|\mathbf{J}_t - \mathbf{J}_{t,t+1}\|_2 \left\|\mathbf{J}_t^\top\right\|_2 \left\|\mathbf{G}_t^{-1}\right\|_2 \|\mathbf{f}_t - \mathbf{y}\|_2.
\end{aligned}
\tag{17}
$$

Then we control the first term in Eq. (17) in a way similar to the following:

$$\|\mathbf{J}_t - \mathbf{J}_{t,t+1}\|_2 \leq \frac{\hat{C}}{\sqrt{M}}\|\mathbf{W}_t - \mathbf{W}_{t+1}\|_2 \leq \frac{\hat{C}}{\sqrt{M}}\left\|\mathbf{J}_t^\top\right\|_2\left\|\mathbf{G}_t^{-1}\right\|_2\|\mathbf{f}_t - \mathbf{y}\|_2,$$

and if $\left\|\mathbf{J}_t^\top\right\|_2\left\|\mathbf{G}_t^{-1}\right\|_2$ can be upper bounded, we get our result Eq. (14).

For the mini-batch version, similarly we have

$$
\begin{aligned}
\mathbf{f}_{t+1} - \mathbf{y} &= \mathbf{f}_t - \mathbf{y} - \mathbf{J}_{t,t+1}\mathbf{J}_{t,B_t}^\top \mathbf{G}_{t,B_t}^{-1}(\mathbf{f}_{t,B_t} - \mathbf{y}_{B_t}) \\
&= \mathbf{f}_t - \mathbf{y} - \mathbf{J}_{t,t+1}\mathbf{J}_{t,B_t}^\top \tilde{\mathbf{G}}_{t,B_t}^{-1}(\mathbf{f}_t - \mathbf{y}),
\end{aligned}
\tag{18}
$$

where the subscript $B_t$ denotes the sub-matrix/vector corresponding to the batch, and $\tilde{\mathbf{G}}_{t,B_t}^{-1} \in \mathbb{R}^{b \times n}$ is a zero-padded version of $\mathbf{G}_{t,B_t}^{-1}$ to make Eq. (18) hold. Therefore, after one epoch (from the $(tk+1)$-th to the $((t+1)k)$-th update), we have

$$\mathbf{f}_{(t+1)k} - \mathbf{y} = \left(\prod_{t'=(t+1)k-1}^{tk}\left(\mathbf{I} - \mathbf{J}_{t',t'+1}\mathbf{J}_{t',B_{t'}}^\top \tilde{\mathbf{G}}_{t',B_{t'}}^{-1}\right)\right)(\mathbf{f}_{tk} - \mathbf{y}) =: \mathbf{A}_t(\mathbf{f}_{tk} - \mathbf{y}).$$

We will see that the matrix $\mathbf{A}_t$ is close to the matrix $\mathbf{A}$ defined in (15), so it boils down to analyzing the spectral properties of $\mathbf{A}$.

For both theorems, we can compute that as $M$ increases, the norm of the update $\|\mathbf{W}_t - \mathbf{W}_{t+1}\|_F$ does not increase with $M$, so the update is small compared to the Gaussian initialization where $\|\mathbf{W}_0\|_2 = \Theta(\sqrt{M})$. From this we can derive that the matrices $\mathbf{J}, \mathbf{G}$ etc. remain close to their initialization, which makes bounding their norms possible. The full proof is in the appendix. □

In conclusion, the accelerated convergence is related to the local linearity and the stability of the Jacobian and Gram matrix. We emphasize that our theorems serve more as a motivation than a justification of our GGN algorithm, because we expect that GGN works in practice, even under milder situations when $M$ is not as large as the theorem demands or for deep networks with different architectures, and that GGN would still perform much better than first-order methods.

### 3.3 ANALYSIS OF THE PER-ITERATION COMPUTATIONAL COMPLEXITY

We have proved that for sufficiently overparametrized deep neural networks, full-batch GGN has quadratic convergence rate. In this subsection, we analyze the per-iteration computational cost of GGN, and compare it to that of SGD.

For every mini-batch (i.e., iteration), there are two major steps of computation in GGN:

- (A). Forward, and then backpropagate for computing the Jacobian matrix $\mathbf{J}$.
- (B). Use $\mathbf{J}$ to compute the update $\mathbf{J}^\top (\lambda \mathbf{G} + \alpha \mathbf{I})^{-1}(\mathbf{f} - \mathbf{y})$

*We show that the computational complexity of (A) is the same as that of SGD with the same batch size; and the computational complexity of (B) is small compared to (A) for typical networks and batch sizes.* Thus, the per-iteration computation overhead of GGN is very small compared to SGD. Overall, in terms of training time, GGN can be much faster than SGD.

For the computation in step (A), the forward part is just the same as that of SGD. For the backward part, *for every input data*, GGN keeps track of the output's derivative for the *nodes* in the middle of the computational graph. This part is just the same as backpropagation in SGD. What is different is that GGN also, *for every input data*, keeps track of the output's derivative for the *parameters*; while in SGD the derivatives for the parameters are *averaged over a batch of data*. However, it is not difficult to see the computational costs of GGN and SGD are the same.

For the computation in step (B), observe that the size of the Jacobian is $b \times m$ where $b$ is the batch size and $m$ is the number of parameters. The Gram matrix $\mathbf{G}_{t,B_t} = \mathbf{J}_{t,B_t}\mathbf{J}_{t,B_t}^\top$ in our Gram-Gauss-Newton method is of size $b \times b$ and it only requires $O(b^2 m + b^3)$ for computing $\mathbf{G}_{t,B_t}$ and a matrix inverse. Multiplying the two matrices to $\mathbf{f} - \mathbf{y}$ requires even less computation. Overall, the computational cost in step (B) is small compared to that of step (A).

## 4 EXPERIMENTS

Given the theoretical findings above, in this section, we compare our proposed GGN algorithm with several baseline algorithms in real applications. In particular, we mainly study two regression tasks, AFAD-LITE (Niu et al., 2016) and RSNA Bone Age (rsn, 2017).

### 4.1 EXPERIMENTAL SETTING

**AFAD-LITE task** is to predict the age of human from the facial information. The training data of the AFAD-LITE task contains 60k facial images and the corresponding age for each image. We choose ResNet-32 (He et al., 2016) as the base model architecture. During training, all input images are resized to $64 * 64$. We study two variants of the ResNet-32 architecture: ResNet-32 with batch normalization layer (referred to as *ResNetBN*), and ResNet-32 with Fixup initialization (Zhang et al., 2019b) (referred to as *ResNetFixup*). In both settings, we use SGD as our baseline algorithm. In particular, we follow Qian (1999) to use its momentum variant and set the hyper-parameters lr=0.003 and momentum=0.9 determined by selecting the best optimization performance using grid search. Since batch normalization is computed over all samples within a mini-batch, it is not consistent with

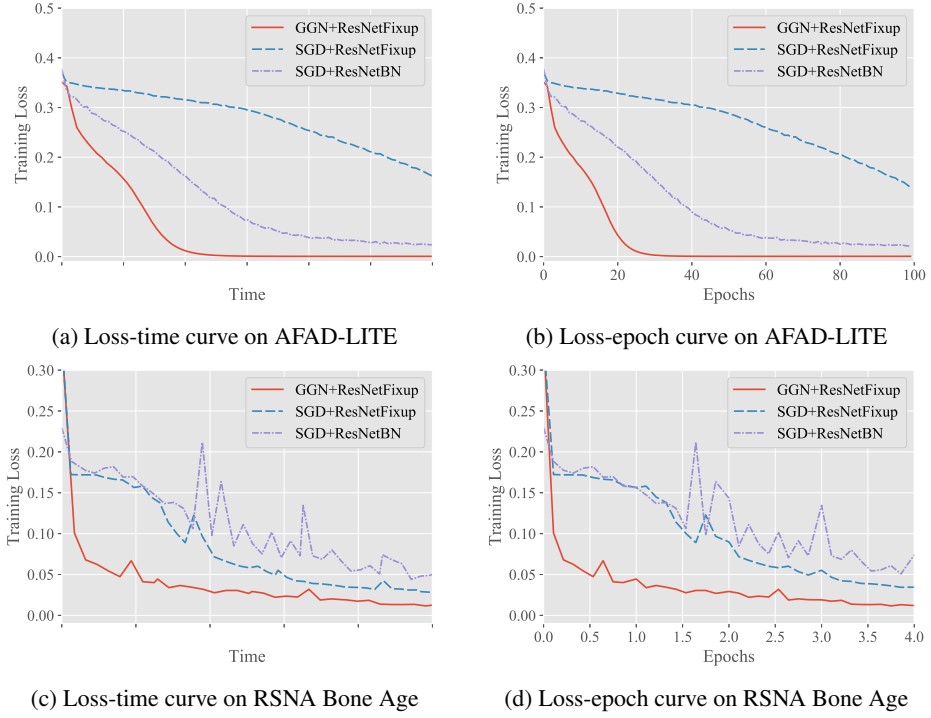

Figure 1: Training curves of GGN and SGD on two regression tasks.

our assumption in Section 2 that the regression function has the form of $f(\mathbf{w}, \mathbf{x})$, which only depends on $\mathbf{w}$ and a single input datum $\mathbf{x}$. For this reason, the GGN algorithm does not directly apply to ResNetBN, and we test our proposed algorithm on ResNetFixup only. We set $\lambda = 1$ and $\alpha = 0.3$ for GGN. We follow the common practice to set the batch size to 128 for our proposed method and all baseline algorithms. Mean square loss is used for training.

**RSNA Bone Age task**  is a part of the 2017 Pediatric Bone Age Challenge organized by the Radiological Society of North America (RSNA). It contains 12,611 labeled images. Each image in this dataset is a radiograph of a left hand labeled with the corresponding bone age. During training, all input images are resized to $64 * 64$. We also choose ResNetBN and ResNetFixup for this experiment, and use ResNetBN and ResNetFixup trained in the first task as warm-start initialization. We use lr= 0.01 and momentum= 0.9 for SGD, and use $\lambda = 1$ and $\alpha = 0.1$ for GGN. Batch size is set to 128 in these experiments, and mean square loss is used for training.

## 4.2 EXPERIMENTAL RESULTS

**Convergence.**  The training loss curves of different optimization algorithms for AFAD-LITE and RSNA Bone Age tasks are shown in Figure 1. On both tasks, our proposed method converges much faster than the baselines. We can see from Figure 1a and Figure 1b that, on the AFAD-LITE task, the loss using our GGN method quickly decreases to nearly zero in 30 epochs. On the contrary, for both baselines using SGD, the loss decays much slower than our method in terms of wall clock time and epochs. Similar advantage of GGN can also be observed on the RSNA bone age task.

**Generalization performance and different hyper-parameters.**  We can see that our proposed method trains much faster than other baselines. However, as a machine learning model, generalization performance also needs to be evaluated. Due to space limitation, we only provide the test curve for the RSNA Bone Age task in Figure 2a. From the figure, we can see that the test loss of our proposed method also decreases faster than the baseline methods. Furthermore, the loss of our GGN algorithm is lower than those of the baselines. These results show that the GGN algorithm can not only accelerate the whole training process, but also learn better models.

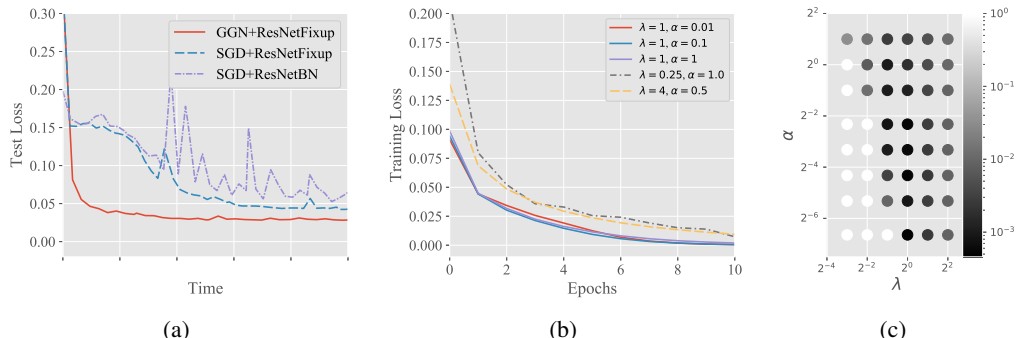

Figure 2: Test performance and ablation study on hyper-parameters on RSNA Bone Age dataset. (a) Test curves of GGN and SGD. (b) Training curves of GGN with different hyper-parameter configurations. The optimal $\alpha$ is shown for $\lambda = 0.25$ and $\lambda = 4$ by grid search. (c) Training loss at $10^{th}$ epoch of models trained using GGN with different hyper-parameters.

We then study the effect of hyper-parameters used in the GGN algorithm. We try different $\lambda$ and $\alpha$ on the RSNA Bone Age task and report the training loss of all experiments at the $10^{th}$ epoch. All results are plotted in Figure 2c. In the figure, the x-axis is the value of $\lambda$ and the y-axis is the value of $\alpha$. The gray value of each point corresponds to the loss, the lighter the color, the higher the loss. We can see that the model converges faster when $\lambda$ is close to 1. In GGN, $\alpha$ can be considered as the inverse value of the learning rate in SGD. Empirically, we find that the convergence speed of training loss is not that sensitive to $\alpha$ given a proper $\lambda$, such as $\lambda = 1$. Some training loss curves of different hyper-parameter configurations are shown in Figure 2b.

## 5 CONCLUSION AND DISCUSSIONS

We propose a novel Gram-Gauss-Newton (GGN) method for solving regression problems with square loss using overparameterized neural networks. Despite being a second-order method, the computation overhead of the GGN algorithm at each iteration is small compared to SGD. We also prove that if the neural network is sufficiently wide, GGN algorithm enjoys a quadratic convergence rate. Experimental results on two regression tasks demonstrate that GGN compares favorably to SGD on these data sets with standard network architectures. Our work illustrates that second-order methods have the potential to compete with first-order methods for learning deep neural networks with huge number of parameters.

In this paper, we mainly focus on the regression task, but our method can be easily generalized to other tasks such as classification as well. Consider the $k$-category classification problem, the neural network outputs a vector with $k$ entries. Although this will increase the computational complexity of getting the Jacobian whose size increases $k$ times, i.e., $\mathbf{J} \in \mathbb{R}^{(bk) \times m}$, each row of $\mathbf{J}$ can be still computed in parallel, which means the extra cost only comes from parallel computation overhead when we calculate in a fully parallel setting. While most first-order methods for training neural networks can hardly make use of the computational resource in parallel or distributed settings to accelerate training, our GGN method can exploit this ability. For first-order methods, basically extra computational resource can only be used to calculate more gradients at a time by increasing batch size, which harms generalization a lot. But for GGN, more resource can be used to refine the gradients and achieve accelerated convergence speed with the help of second-order information. It is an important future work to study the application of GGN to classification problems.

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

## A  SOME PREPARATIONS FOR THE PROOF OF THEOREMS

**Notations.**    We use the following notations for the rest of the sections.

- Let $[n] = \{1, \cdots, n\}$.
- $J_{\mathbf{W},\mathbf{x}} \in \mathbb{R}^{M \times d}$ denotes the gradient $\frac{\partial f(\mathbf{W},\mathbf{x})}{\partial \mathbf{W}}$, which is of the same size of $\mathbf{W}$.
- The bold $\mathbf{J_W}$ or $\mathbf{J}(\mathbf{W})$ denotes the Jacobian with regard to all $n$ data, with each gradient for $\mathbf{W}$ vectorized, i.e.

$$\mathbf{J_W} = \begin{bmatrix} \mathrm{vec}(J_{\mathbf{W},\mathbf{x_1}})^\top \\ \vdots \\ \mathrm{vec}(J_{\mathbf{W},\mathbf{x}_n})^\top \end{bmatrix} \in \mathbb{R}^{n \times (Md)}. \qquad (19)$$

- $\mathbf{w}_r$ denotes the $r$-th row of $\mathbf{W}$, which is the incoming weights of the $r$-th neuron.
- $\mathbf{W}_0$ stands for the parameters at initialization.
- $\mathbf{d}_{\mathbf{W},\mathbf{x}} := \sigma'(\mathbf{Wx}) \in \mathbb{R}^{M \times 1}$ denotes the (entry-wise) derivative of the activation function.
- We use $\langle \cdot, \cdot \rangle$ to denote the inner product, $\|\cdot\|_2$ to denote the Euclidean norm for vectors or the spectral norm for matrices, and $\|\cdot\|_F$ to denote the Frobenius norm for matrices.

We can easily derive the formula for $J$ as

$$J_{\mathbf{W},\mathbf{x}} = \frac{1}{\sqrt{M}}((\mathbf{d}_{\mathbf{W},\mathbf{x}} \circ \mathbf{a})\mathbf{x}^\top), \tag{20}$$

where $\circ$ is the point-wise product. So we can also easily solve $G$ as

$$\mathbf{G}_{ij} = \langle J_{\mathbf{x}_i}, J_{\mathbf{x}_j} \rangle = \frac{1}{M} \sum_{r=1}^{M} \mathbf{x}_i^\top \mathbf{x}_j \sigma'(\mathbf{w}_r \mathbf{x}_i)\sigma'(\mathbf{w}_r \mathbf{x}_j). \tag{21}$$

Our analysis is based on the fact that $\mathbf{G}$ stays not too far from its infinite-width limit

$$\mathbf{K}(\mathbf{x}_i, \mathbf{x}_j) = \mathbb{E}_{\mathbf{w} \sim \mathcal{N}(0, \mathbf{I}_d)} \left( \mathbf{x}_i^\top \mathbf{x}_j \sigma'(\mathbf{w}\mathbf{x}_i)\sigma'(\mathbf{w}\mathbf{x}_j) \right),$$

which is a positive definite matrix with least eigenvalue denoted $\lambda_0$, and we assume $\lambda_0 > 0$. $\lambda_0$ is a small data-dependent constant, and without loss of generality we assume $\lambda_0 \le 1$, or else we can just take $\lambda_0 = 1$ if $\lambda_{\min}(\mathbf{K}) > 1$.

The first lemma is about the estimation of relevant norms at initialization.

**Lemma 2** (Bounds on Norms at Initialization). *If $M = \Omega\left(d \log(16n/\delta)\right)$, then with probability at least $1 - \delta/2$ the following holds*

*(a).* $\|\mathbf{W}_0\|_2 = O(\sqrt{M})$.

*(b).* $f(\mathbf{W}_0, \mathbf{x}_i) = O(1)$, *for $i \in [n]$.*

*(c).* $\|J_{\mathbf{W}_0, \mathbf{x}_i}\|_F = O(1)$, *for $i \in [n]$.*

*Proof.* (a). The lemma is a well-known result concerning the estimation of singular values of Gaussian random matrices (see Corollary 5.35 of Vershynin (2010)). Notice that $\mathbf{W}_0 \in \mathbb{R}^{M \times d}$ is a Gaussian random matrix, the Corollary states that with probability $1 - 2e^{-t^2/2}$ one has

$$\|\mathbf{W}_0\|_2 \le \sqrt{M} + \sqrt{d} + t.$$

By choosing $M = \max(d, \sqrt{2 \log(8/\delta)})$, we obtain $\|\mathbf{W}_0\|_2 \le 3\sqrt{M}$ with probability $1 - \delta/4$.

(b). First, $\mathbf{a}_r, r \in [M]$ are Rademacher variables, thereby 1-sub-Gaussian, so with probability $1 - 2e^{-Mt^2/2}$ we have $\frac{1}{M} \sum_{r=1}^{M} \mathbf{a}_r \le t$. This means if we take $M = \Omega\left(\log(16/\delta)\right)$,

$$\Pr[\frac{1}{\sqrt{M}} \sum_{r=1}^{M} \mathbf{a}_r = O(1)] \ge 1 - \frac{\delta}{8}. \tag{22}$$

Next, the vector $\mathbf{v}_i = \frac{1}{\|\mathbf{x}_i\|_2} \mathbf{W}_0 \mathbf{x}_i \in \mathbb{R}^{M \times 1}$ is a standard Gaussian vector. Suppose the activation $\sigma(\cdot)$ is $l$-Lipschitz and $l$ is $O(1)$ by our assumption, with the vector $\mathbf{a}$ fixed, the function

$$\phi : \mathbb{R}^M \to \mathbb{R}, \mathbf{v}_i \mapsto \frac{1}{\sqrt{M}} \mathbf{a}^\top \sigma(\|\mathbf{x}_i\|_2 \mathbf{v}_i) = f(\mathbf{W}_0, \mathbf{x}_i)$$

has a Lipschitz parameter of $l \|\mathbf{x}_i\|_2 / \sqrt{M} = O(1/\sqrt{M})$. According to the classic result on the concentration of a Lipschitz function over Gaussian variables (see Theorem 2.26 of Wainwright (2019)), we have

$$\Pr[|\phi(\mathbf{v}_i) - \mathbb{E}_{\mathbf{W}_0}(\phi(\mathbf{v}_i))| \ge t] \le 2 \exp\left(-\frac{Mt^2}{2l^2 \|\mathbf{x}_i\|^2}\right),$$

which means if $M = \Omega(\log(16n/\delta))$,

$$\left| \frac{1}{\sqrt{M}} \mathbf{a}^\top \sigma(\mathbf{W}_0 \mathbf{x}_i) - \frac{1}{\sqrt{M}} (\sum_{r=1}^{M} \mathbf{a}_r) \mathbb{E}_{\mathbf{w} \sim \mathcal{N}(0, \mathbb{I}_d)}[\sigma(\mathbf{w}\mathbf{x}_i)] \right| = O(1) \tag{23}$$

holds jointly for all $i \in [n]$ with probability $1 - \delta/8$. Note that

$$\left| \mathbb{E}_{\mathbf{w} \sim \mathcal{N}(0, \mathbb{I}_d)}[\sigma(\mathbf{w}\mathbf{x}_i)] \right| \le |\sigma(0)| + l \cdot \mathbb{E}_{\xi \sim \mathcal{N}(0, \|\mathbf{x}_i\|_2^2))}[|\xi|] = O(1). \tag{24}$$

Plugging in (22) and (24) into (23), we see that as long as $M = \Omega(\log(16n/\delta))$, then with probability $1 - \delta/4$, for all $i \in [n]$,

$$f(\mathbf{W}_0, \mathbf{x}_i) = \frac{1}{\sqrt{M}} \mathbf{a}^\top \sigma(\mathbf{W}_0 \mathbf{x}_i) = O(1).$$

(c). Since $\sigma$ is $O(1)$-Lipschitz, we have $\|\mathbf{d}_{\mathbf{W}, \mathbf{x}_i}\|_\infty = O(1)$. According to (20) we can easily know that $\|J_{\mathbf{W}, \mathbf{x}_i}\|_F \leq \frac{1}{\sqrt{M}} \|\text{Diag}(\mathbf{d})\|_2 \|\mathbf{a}\|_2 \|\mathbf{x}\|_2 = O(1)$. □

The next lemma is about the least eigenvalue of the Gram matrix $G$ at initialization. It shows that when $M$ is large, $\mathbf{G}_{\mathbf{W}_0}$ is close to $\mathbf{K}$ and has a lower bounded least eigenvalue. It is the same as Lemma 3.1 in Du et al. (2018b), but here we restate it and its proof for the reader's convenience.

**Lemma 3** (Bound on the Least Eigenvalue of the Gram Matrix at Initialization). *If the width* $M = \Omega\left(\frac{n^2 \log(2n/\delta)}{\lambda_0^2}\right)$, *then with probability at least* $1 - \delta/2$ *over random initialization, we have*

$$\lambda_{\min}(\mathbf{G}_{\mathbf{W}_0}) \geq \frac{3}{4}\lambda_0.$$

*Proof.* Because $\sigma$ is Lipschitz, $\sigma'(\mathbf{w}\mathbf{x}_i)\sigma'(\mathbf{w}\mathbf{x}_j)$ is bounded by $O(1)$. For every fixed $(i, j)$ pair, at initialization $\mathbf{G}_{ij}$ is an average of independent random variables, and by Hoeffding's inequality, applying union bound for all $n^2$ of $(i, j)$ pairs, with probability $1 - \delta/2$ at initialization we have

$$|\mathbf{G}_{ij} - \mathbf{K}_{ij}| \leq O\left(\sqrt{\frac{\log(2n/\delta)}{M}}\right)$$

and then

$$\|\mathbf{G}_{\mathbf{W}_0} - \mathbf{K}\|_2^2 \leq \|\mathbf{G}_{\mathbf{W}_0} - \mathbf{K}\|_F^2 = O\left(\frac{n^2 \log(2n/\delta)}{M}\right).$$

Thus if $M = \Omega\left(\frac{n^2 \log(2n/\delta)}{\lambda_0^2}\right)$ we can have $\|\mathbf{G}_{\mathbf{W}_0} - \mathbf{K}\|_2 \leq \frac{1}{4}\lambda_0$ and thus $\lambda_{\min}(\mathbf{G}_{\mathbf{W}_0}) \geq \frac{3}{4}\lambda_0$. □

Next, in Lemma 4 and 5 we will bound the relevant norms and the least eigenvalue of $\mathbf{G}$ inside some scope of $\mathbf{W}$ that covers the whole optimization trajectory starting from $\mathbf{W}_0$. Specifically, we consider the range

$$B(R) \triangleq \{\mathbf{W} : \|\mathbf{W} - \mathbf{W}_0\|_F \leq R\},$$

where $R$ is determined later to make sure that the optimization trajectory remains inside $B(R)$. The idea of the whole convergence theorem is that when the width $M$ is large, $R$ is very small compared to its initialization scale: $\|\mathbf{W}_0\|_2 = O(\sqrt{M})$. This way, neither the Jacobian nor the Gram matrix changes much during optimization.

**Lemma 4** (Bounds on Norms in the Optimization Scope). *Suppose the events in Lemma 2 hold. There exists a constant $C > 0$ such that if $M \geq CR^2$, we have the following:*

*(a) For any $\mathbf{W} \in B(R)$, we have*

$$\|\mathbf{W}\|_2 = O(\sqrt{M}). \tag{25}$$

*(b) For any $\mathbf{W}_1, \mathbf{W}_2 \in B(R)$, if $\|\mathbf{W}_1 - \mathbf{W}_2\|_F \leq R'$, then we have*

$$\|J_{\mathbf{W}_1, \mathbf{x}_i} - J_{\mathbf{W}_2, \mathbf{x}_i}\|_F = O\left(\frac{R'}{\sqrt{M}}\right), \ \forall i, \text{ and } \|\mathbf{J}_{\mathbf{W}_1} - \mathbf{J}_{\mathbf{W}_2}\|_2 = O\left(R'\sqrt{\frac{n}{M}}\right). \tag{26}$$

*Also, for any $\mathbf{W} \in B(R)$, we have*

$$\|J_{\mathbf{W}}\|_F = O(1) \text{ and } \|\mathbf{J}_{\mathbf{W}}\|_2 = O(\sqrt{n}). \tag{27}$$

*Proof.* (a). This is straightforward from Lemma 2(a), the definition of $B(R)$, and $M = \Omega(R^2)$.

(b). According to the $O(1)$-smoothness of the activation, we have

$$\|\mathbf{d}_{\mathbf{W}_1,\mathbf{x}_i} - \mathbf{d}_{\mathbf{W}_2,\mathbf{x}_i}\|_2 \leq O(1) \cdot \|\mathbf{W}_1 \mathbf{x}_i - \mathbf{W}_2 \mathbf{x}_i\|_2 = O(R'),$$

so we can bound

$$
\begin{aligned}
&\|J_{\mathbf{W}_1,\mathbf{x}_i} - J_{\mathbf{W}_2,\mathbf{x}_i}\|_F \\
=& \frac{1}{\sqrt{M}} \left\| \mathrm{Diag}(\mathbf{a})(\mathbf{d}_{\mathbf{W}_1,\mathbf{x}_i} - \mathbf{d}_{\mathbf{W}_2,\mathbf{x}_i})\mathbf{x}^\top \right\|_F \\
\leq& \frac{1}{\sqrt{M}} \left\| \mathrm{Diag}(\mathbf{a}) \right\|_2 \left\| \mathbf{d}_{\mathbf{W}_1,\mathbf{x}_i} - \mathbf{d}_{\mathbf{W}_2,\mathbf{x}_i} \right\|_2 \|\mathbf{x}_i\|_2 \\
\leq& O(\frac{R'}{\sqrt{M}}).
\end{aligned}
$$

And according to (19), we have

$$
\begin{aligned}
\|\mathbf{J}_{\mathbf{W}_1} - \mathbf{J}_{\mathbf{W}_2}\|_2 &\leq \|\mathbf{J}_{\mathbf{W}_1} - \mathbf{J}_{\mathbf{W}_2}\|_F \\
&= \sqrt{\sum_{i=1}^n \|J_{\mathbf{W}_1,\mathbf{x}_i} - J_{\mathbf{W}_2,\mathbf{x}_i}\|_F^2} \\
&= O\left(R'\sqrt{\frac{n}{M}}\right).
\end{aligned}
$$

Also, taking $\mathbf{W}_1 = \mathbf{W}$ and $\mathbf{W}_2 = \mathbf{W}_0$, combining with Lemma 2(c), we see there exists $C$ such that for $M \geq CR^2$ we have $\|\mathbf{J}_{\mathbf{W}}\|_F = O(1)$, and naturally $\|\mathbf{J}_{\mathbf{W}}\|_2 = O(\sqrt{n})$. Note: The constants hidden in the $O(\cdot)$ notation are irrelevant with $C$. $\qquad\square$

The next Lemma deals with the Gram matrix within $B(R)$, ensuring a lower bound on the least eigenvalue throughout the optimization process (Du et al., 2018a).

**Lemma 5** (Least Eigenvalue in the Optimization Scope). *For $\mathbf{W} \in B(R)$, suppose the events in Lemma 2 and 3 hold, there exists a constant $C'$ such that for $M \geq \frac{C'n^2 R^2}{\lambda_0^2}$, we have*

$$\|\mathbf{G}_{\mathbf{W}} - \mathbf{G}_{\mathbf{W}_0}\|_2 \leq \frac{\lambda_0}{4},$$

*and thus combined with Lemma 3, we know that $\mathbf{G}_{\mathbf{W}}$ remains invertible when $\mathbf{W} \in B(R)$ and satisfies $\|\mathbf{G}_{\mathbf{W}}^{-1}\|_2 \leq \frac{2}{\lambda_0}$.*

*Proof.* Based on the results in Lemma 4(b), we have

$$\|\mathbf{G}_{\mathbf{W}} - \mathbf{G}_{\mathbf{W}_0}\|_2 = \left\|\mathbf{J}_{\mathbf{W}}\mathbf{J}_{\mathbf{W}}^\top - \mathbf{J}_{\mathbf{W}_0}\mathbf{J}_{\mathbf{W}_0}^\top\right\|_2 \tag{28}$$

$$\leq \left\|(\mathbf{J}_{\mathbf{W}} - \mathbf{J}_{\mathbf{W}_0})\mathbf{J}_{\mathbf{W}}^\top)\right\|_2 + \left\|\mathbf{J}_{\mathbf{W}_0}(\mathbf{J}_{\mathbf{W}}^\top - \mathbf{J}_{\mathbf{W}_0}^\top)\right\|_2 \tag{29}$$

$$\leq O(\frac{nR}{\sqrt{M}}). \tag{30}$$

To make the above less than $\frac{\lambda_0}{4}$, choosing $M$ greater than some $\frac{C'n^2 R^2}{\lambda_0^2}$ suffices, and the lemma is thus proved. Again the constants hidden in the $O(\cdot)$ notation are irrelevant with $C'$. $\qquad\square$

## B  PROOF OF THEOREM 1

**Proof idea.** In this section, we use $\mathbf{W}_t, t \in \{0, 1, \cdots\}$ to represent the parameter $\mathbf{W}$ after $t$ iterations. For convenience, $\mathbf{J}_t, \mathbf{G}_t, \mathbf{f}_t$ is short for $\mathbf{J}_{\mathbf{W}_t}, \mathbf{G}_{\mathbf{W}_t}, \mathbf{f}_{\mathbf{W}_t}$ respectively. We introduce

$$\mathbf{J}_{t,t+1} = \int_0^1 \mathbf{J}((1-s)\mathbf{W}_t + s\mathbf{W}_{t+1})ds. \tag{31}$$

For each iteration, if $\mathbf{G}_t$ is invertible, we have

$$
\begin{aligned}
\mathbf{f}_{t+1} - \mathbf{y} &= \mathbf{f}_t - \mathbf{y} + \mathbf{J}_{t,t+1} \operatorname{vec}(\mathbf{W}_{t+1} - \mathbf{W}_t) \\
&= \mathbf{f}_t - \mathbf{y} - \mathbf{J}_{t,t+1} \mathbf{J}_t^\top \mathbf{G}_t^{-1} (\mathbf{f}_t - \mathbf{y}) \\
&= (\mathbf{J}_t - \mathbf{J}_{t,t+1}) \mathbf{J}_t^\top \mathbf{G}_t^{-1} (\mathbf{f}_t - \mathbf{y}),
\end{aligned}
$$

hence

$$
\|\mathbf{f}_{t+1} - \mathbf{y}\|_2 \leq \|\mathbf{J}_t - \mathbf{J}_{t,t+1}\|_2 \left\|\mathbf{J}_t^\top\right\|_2 \left\|\mathbf{G}_t^{-1}\right\|_2 \|\mathbf{f}_t - \mathbf{y}\|_2 . \tag{32}
$$

Then we control the first term of the right hand side based on Lemma 4 in the following form

$$
\begin{aligned}
\|\mathbf{J}_t - \mathbf{J}_{t,t+1}\|_2 &\leq \frac{\hat{C}}{\sqrt{M}} \|\mathbf{W}_t - \mathbf{W}_{t+1}\|_2 \\
&= \frac{\hat{C}}{\sqrt{M}} \left\|\mathbf{J}_t^\top \mathbf{G}_t^{-1}(\mathbf{f}_t - \mathbf{y})\right\|_2 \\
&\leq \frac{\hat{C}}{\sqrt{M}} \left\|\mathbf{J}_t^\top\right\|_2 \left\|\mathbf{G}_t^{-1}\right\|_2 \|\mathbf{f}_t - \mathbf{y}\|_2 ,
\end{aligned}
$$

and plugging into (32) along with norm bounds on $\mathbf{J}$ and $\mathbf{G}$ we obtain a second-order convergence.

**Formal proof.** Let $R_t = \|\mathbf{W}_t - \mathbf{W}_{t+1}\|_F$ for $t \in \{0, 1, \cdots\}$. We take $R = \Theta\left(\frac{n}{\lambda_0}\right)$ in Lemma 4 and 5 (the constant is chosen to make the right hand side of (34) hold). We prove that there exists an $M = \Omega\left(\max\left(\frac{n^4}{\lambda_0^4}, \frac{n^2 d \log(16n/\delta)}{\lambda_0^2}\right)\right)$ (with enough constant) that suffices. First we can easily verify that all the requirements for $M$ in Lemma 2-5 can be satisfied . Hence, with probability at least $1 - \delta$ all the events in Lemma 2-5 hold. Under this situation, we do induction on $t$ to show the following:

- (a). $\mathbf{W}_t \in B(R)$.
- (b). If $t > 0$, then $\|\mathbf{f}_t - \mathbf{y}\|_2 \leq \frac{n^{3/2}}{\lambda_0^2 \sqrt{M}} \|\mathbf{f}_{t-1} - \mathbf{y}\|_2^2$

As long as (b) is true for all $t$, then choosing $M$ large enough to make sure the series $\{\|\mathbf{f}_t - \mathbf{y}\|_2\}_{t=0}^\infty$ converges to zero, we obtain the second-order convergence property.

For $t = 0$, (a) and (b) hold by definition. Suppose the proposition holds for $t = 0, \cdots, T$. Then for $t = 0, \cdots, T$, $\mathbf{G}_t$ is invertible. Recall that the update rule is $\operatorname{vec}(\mathbf{W}_{t+1}) = \operatorname{vec}(\mathbf{W}_t) - \mathbf{J}_t^\top \mathbf{G}_t^{-1}(\mathbf{f}_t - \mathbf{y})$, we have

$$
\begin{aligned}
R_t &= \|\mathbf{W}_t - \mathbf{W}_{t+1}\|_F \\
&= \|\operatorname{vec}(\mathbf{W}_t) - \operatorname{vec}(\mathbf{W}_{t+1})\|_2 \\
&\leq \left\|\mathbf{J}_t^\top\right\|_2 \left\|\mathbf{G}_t^{-1}\right\|_2 \|\mathbf{f}_t - \mathbf{y}\|_2 \\
&\leq O\left(\frac{\sqrt{n}}{\lambda_0} \|\mathbf{f}_t - \mathbf{y}\|_2\right). \qquad \text{(Lemma 4 and 5)}
\end{aligned} \tag{33}
$$

According to Lemma 2(b) and the assumption that the target label $y_i = O(1)$, we have $\|\mathbf{f}_0 - \mathbf{y}\|_2^2 = O(n)$. When $T > 1$, the decay ratio at the first step is bounded as

$$
\frac{\|\mathbf{f}_1 - \mathbf{y}\|_2}{\|\mathbf{f}_0 - \mathbf{y}\|_2} \leq O\left(\frac{n^{\frac{3}{2}}}{\lambda_0^2 \sqrt{M}}\right) \|\mathbf{f}_0 - \mathbf{y}\|_2 \leq O\left(\frac{n^2}{\lambda_0^2 \sqrt{M}}\right) =: r,
$$

and taking $M = \Omega(\frac{n^4}{\lambda_0^4})$ with enough constant can make sure $r$ is a constant less than 1, in which case the second-order convergence property (b) will ensure a faster ratio of decay at each subsequent step in $\|\mathbf{f}_t - \mathbf{y}\|_{t=0}^T$. Combining (33), we have

$$
\sum_{t=0}^T R_t \leq O\left(\sum_{t=0}^T \frac{\sqrt{n}}{\lambda_0} \|\mathbf{f}_t - \mathbf{y}\|_2\right) \leq O\left(\frac{n}{\lambda_0} \sum_{t=1}^\infty r^{t-1}\right) = O\left(\frac{n}{\lambda_0}\right) \leq R. \tag{34}
$$

Therefore, $\mathbf{W}_{T+1} \in B(R)$, and (34) also holds when $T = 0$, so (a) is true.

Since $B(R)$ is convex, this means we also have $s\mathbf{W}_T + (1-s)\mathbf{W}_{T+1} \in B(R)$ for $s \in [0,1]$. Hence we can bound the difference of the Jacobian as

$$\|\mathbf{J}_{T,T+1} - \mathbf{J}_T\|_2 \leq \int_0^1 \|\mathbf{J}(s\mathbf{W}_T + (1-s)\mathbf{W}_{T+1}) - \mathbf{J}(\mathbf{W}_T)\|_2 \, ds$$

$$\leq O\left(\int_0^1 sR_t\sqrt{\frac{n}{M}} ds\right) \qquad \text{(Lemma 4(b))}$$

$$= O\left(\frac{n}{\lambda_0\sqrt{M}} \|\mathbf{f}_T - \mathbf{y}\|_2\right). \qquad \text{(Using (33))} \qquad (35)$$

So we use the bound of (32) and obtain

$$\|\mathbf{f}_{T+1} - \mathbf{y}\|_2 \leq \|\mathbf{J}_T - \mathbf{J}_{T,T+1}\|_2 \|\mathbf{J}_T^\top\|_2 \|\mathbf{G}_T^{-1}\|_2 \|\mathbf{f}_T - \mathbf{y}\|$$

$$\leq \frac{n^{3/2}}{\lambda_0^2\sqrt{M}} \|\mathbf{f}_T - \mathbf{y}\|_2^2, \qquad \text{(Using (35) and Lemma 4, 5)}$$

which proves (b). This concludes our proof.

## C  PROOF OF THEOREM 2

In this section, we give the proof of the convergence of our mini-batch GGN algorithm (Theorem 2).

**Some additional notations.** For the convenience of our proof, we will use slightly different notations than that in Section 3.2. Let $n = bk$, where $b$ is the batch size, $k$ is the number of batches, or equivalently, number of updates in each epoch. For epoch $t \in \{1, 2, \cdots\}$ and batch $i \in [k]$, we will use $\mathbf{W}_{ti} \in \mathbb{R}^{d \times M}$ to denote the parameters before the $i$-th update in epoch $t$. Let $\mathbf{J}_{ti} = \mathbf{J}(\mathbf{W}_{ti}) \in \mathbb{R}^{n \times Md}$, $\mathbf{G}_{ti} = \mathbf{G}(\mathbf{W}_{ti}) \in \mathbb{R}^{n \times n}$, and $\mathbf{f}_{ti} = \mathbf{f}(\mathbf{W}_{ti}) \in \mathbb{R}^{n \times 1}$. (All of $\mathbf{W}_0, \mathbf{W}_1, \mathbf{W}_{11}, \mathbf{G}_0$, etc. can represent initial values in the proof.) Also, let $\mathbf{W}_t = \mathbf{W}_{t1}, \mathbf{J}_t = \mathbf{G}_{t1}, \mathbf{G}_t = \mathbf{J}_{t1}, \mathbf{f}_t = \mathbf{f}_{t1}$, and let $\mathbf{W}_{t(k+1)} = \mathbf{W}_{(t+1)1}, \mathbf{J}_{t(k+1)} = \mathbf{J}_{(t+1)1}$, etc. In the $i$-th batch, we use the data $(\mathbf{x}_l, y_l)_{l=(i-1)b+1}^{ib}$. To specify the matrix related to a batch, we use the following indexing method: For $i', i'' \in [k]$, let

$$\mathbf{J}_{ti,i'} = \mathbf{J}_{ti}((i'-1)b + 1 : i'b, 1 : Md) \in \mathbb{R}^{b \times (Md)},$$

which is the $b$ rows of $\mathbf{J}_{ti}$ that represent the Jacobian of the $i$-th batch. Similarly, we have

$$\mathbf{G}_{ti,i'i''} = \mathbf{J}_{ti,i'}\mathbf{J}_{ti,i''}^\top \in \mathbb{R}^{b \times b},$$

$$\mathbf{f}_{ti,i'} = \mathbf{f}_{ti}((i'-1)b + 1 : ib) \in \mathbb{R}^{b \times 1},$$

$$\mathbf{y}_{i'} = \mathbf{y}_{((i'-1)b+1:ib)} \in \mathbb{R}^{b \times 1}.$$

Similar to (31) in the proof of Theorem 1, we make use of the notation

$$\mathbf{J}_{t(i,i+1)} = \int_0^1 \mathbf{J}(s\mathbf{W}_{ti} + (1-s)\mathbf{W}_{t(i+1)})ds, \qquad (36)$$

and similarly define $\mathbf{J}_{t(i,i+1),i'} \in \mathbb{R}^{b \times Md}$, etc. In addition, we make use of the matrix

$$\tilde{\mathbf{G}}_{ti} = \begin{bmatrix} \mathbf{0}_{b \times (i-1)b} & \mathbf{G}_{ti,ii}^{-1} & \mathbf{0}_{b \times (k-i)b} \end{bmatrix} \in \mathbb{R}^{b \times n}$$

in the formula of the update.

Similar to the proof of Theorem 1, the idea of our proof is that due to the fact that the change of $\mathbf{W}$ is small compared to the scale of its initialization, the matrix $\mathbf{J}$ and $\mathbf{G}$ remains stable. Informally, this makes the algorithm close to solving kernel regression iteratively by batches, which is equivalent to solving a system of linear equations

$$\mathbf{J}\mathbf{J}^\top \mathbf{r} = \mathbf{f}(\mathbf{W}_0) - \mathbf{y}, \qquad (37)$$

for variables $\mathbf{r} \in \mathbb{R}^{n \times 1}$ (where $\text{vec}(\mathbf{W}) = \text{vec}(\mathbf{W}_0) + \mathbf{J}^\top \mathbf{r}$), using the Gauss-Siedel method, which means solving

$$\mathbf{J}\mathbf{J}^\top \mathbf{G} \begin{bmatrix} \mathbf{r}^{(\text{old})}(1:(i-1)b) \\ \mathbf{r}^{(\text{new})}((i-1)b+1:ib) \\ \mathbf{r}^{(\text{old})}((ib+1:kb)) \end{bmatrix} = (\mathbf{f}(\mathbf{W}_0 - \mathbf{y}))_{(i-1)b+1:ib}$$

for the $i$-th batch in every epoch. Therefore, it is natural that the matrix $\mathbf{A} = \mathbf{L}^\top(\mathbf{D} - \mathbf{L})^{-1}$ in (15) is introduced. We will show later that the real update follows

$$\mathbf{f}_{t+1} - \mathbf{y}_t = \mathbf{A}_t(\mathbf{f}_t - \mathbf{y})$$

for some $\mathbf{A}_t \approx \mathbf{A}$.

In order to prove the theorem, we need some additional lemmas as follows.

**Lemma 6** (Formula for the Update). *If the Gram matrix at each step is invertible, then:*

*(a) The update of $\mathbf{W}$ is*

$$\text{vec}(\mathbf{W}_{t(i+1)}) = \text{vec}(\mathbf{W}_{ti}) - \mathbf{J}_{ti,i}^\top \mathbf{G}_{ti,ii}^{-1}(\mathbf{f}_{ti,i} - \mathbf{y}_i) \tag{38}$$

$$= \text{vec}(\mathbf{W}_{ti}) - \mathbf{J}_{ti,i}^\top \tilde{\mathbf{G}}_{ti}(\mathbf{f}_t - \mathbf{y}) \tag{39}$$

*(b) The formula for $\mathbf{f} - \mathbf{y}$ is*

$$\mathbf{f}_{t(i+1)} - \mathbf{y} = (\mathbf{I}_n - \mathbf{J}_{t(i,i+1)}\mathbf{J}_{ti,i}^\top \tilde{\mathbf{G}}_{ti})(\mathbf{f}_{ti} - \mathbf{y}) \tag{40}$$

*(c) The update of $\mathbf{f} - \mathbf{y}$ satisfies*

$$\mathbf{f}_{t(i+1)} - \mathbf{y} = \mathbf{U}_{ti}(\mathbf{D}_t - \mathbf{L}_t)^{-1}(\mathbf{f}_t - \mathbf{y}), \tag{41}$$

*where*

$$\mathbf{D}_t = \begin{bmatrix} \mathbf{G}_{t1,11} & \mathbf{0} & \cdots & \mathbf{0} \\ \mathbf{0} & \mathbf{G}_{t2,22} & \cdots & \mathbf{0} \\ \vdots & \vdots & \ddots & \vdots \\ \mathbf{0} & \mathbf{0} & \cdots & \mathbf{G}_{tk,kk} \end{bmatrix}, \quad \mathbf{L}_t = -\begin{bmatrix} \mathbf{0} & \mathbf{0} & \cdots & \mathbf{0} \\ \mathbf{J}_{t(1,2),2}\mathbf{J}_{t1,1}^\top & \mathbf{0} & \cdots & \mathbf{0} \\ \vdots & \vdots & \ddots & \vdots \\ \mathbf{J}_{t(1,2),k}\mathbf{J}_{t1,1}^\top & \mathbf{J}_{t(2,3),k}\mathbf{J}_{t2,2}^\top & \cdots & \mathbf{0} \end{bmatrix},$$

*and*

$$\mathbf{U}_{ti} = \text{Diag}\left( -\begin{bmatrix} \mathbf{J}_{t(1,2),1}\mathbf{J}_{t1,1}^\top - \mathbf{G}_{t1,11} & \mathbf{J}_{t(2,3),1}\mathbf{J}_{t2,2}^\top & \cdots & \mathbf{J}_{t(i,i+1),1}\mathbf{J}_{ti,i}^\top \\ \mathbf{0} & \mathbf{J}_{t(2,3),2}\mathbf{J}_{t2,2}^\top - \mathbf{G}_{t2,22} & \cdots & \mathbf{J}_{t(i,i+1),2}\mathbf{J}_{ti,i}^\top \\ \vdots & \vdots & \ddots & \vdots \\ \mathbf{0} & \mathbf{0} & \cdots & \mathbf{J}_{t(i,i+1),i}\mathbf{J}_{ti,i}^\top - \mathbf{G}_{ti,ii} \end{bmatrix}_{ib \times ib} \right.$$

$$\left. \begin{bmatrix} \mathbf{G}_{t(i+1),(i+1)(i+1)} & \mathbf{0} & \cdots & \mathbf{0} \\ \mathbf{J}_{t(i+1,i+2),i+2}\mathbf{J}_{t(i+1),i+1}^\top & \mathbf{G}_{t(i+2),(i+2)(i+2)} & \cdots & \mathbf{0} \\ \vdots & \vdots & \ddots & \vdots \\ \mathbf{J}_{t(i+1,i+2),k}\mathbf{J}_{t(i+1),i+1}^\top & \mathbf{J}_{t(i+2,i+3),k}\mathbf{J}_{t(i+2),i+2}^\top & \cdots & \mathbf{G}_{tk,kk} \end{bmatrix}_{(k-i)b \times (k-i)b} \right).$$

*Or, if we denote*

$$\mathbf{U}_{ti} = \begin{bmatrix} \tilde{\mathbf{U}}_{ti} \ (\in \mathbb{R}^{ib \times ib}) & \mathbf{0} \\ \mathbf{0} & \overline{\mathbf{U}}_{ti}(\in \mathbb{R}^{(k-i)b \times (k-i)b}) \end{bmatrix},$$

$$\mathbf{D}_{ti} - \mathbf{L}_{ti} = \begin{bmatrix} \tilde{\mathbf{D}}_{ti} - \tilde{\mathbf{L}}_{ti} \ (\in \mathbb{R}^{ib \times ib}) & \mathbf{0} \\ \hat{\mathbf{D}}_{ti} - \hat{\mathbf{L}}_{ti} & \overline{\mathbf{D}}_{ti} - \overline{\mathbf{L}}_{ti}(\in \mathbb{R}^{(k-i)b \times (k-i)b}) \end{bmatrix},$$

*then we have*

$$\mathbf{f}_{t(i+1)} - \mathbf{y} = \begin{bmatrix} \tilde{\mathbf{U}}_{ti}(\tilde{\mathbf{D}}_{ti} - \tilde{\mathbf{L}}_{ti})^{-1} & \mathbf{0} \\ -(\hat{\mathbf{D}}_{ti} - \hat{\mathbf{L}}_{ti})(\tilde{\mathbf{D}}_{ti} - \tilde{\mathbf{L}}_{ti})^{-1} & \mathbf{I}_{(k-i)b} \end{bmatrix}(\mathbf{f}_t - \mathbf{y}) \tag{42}$$

*Specifically, we have*

$$\mathbf{f}_{t+1} - \mathbf{y} = \mathbf{U}_t(\mathbf{D}_t - \mathbf{L}_t)^{-1}(\mathbf{f}_t - \mathbf{y}) =: \mathbf{A}_t(\mathbf{f}_t - \mathbf{y}), \tag{43}$$

*where*

$$\mathbf{U}_t = - \begin{bmatrix} \mathbf{J}_{t(1,2),1}\mathbf{J}_{t1,1}^\top - \mathbf{G}_{t1,11} & \mathbf{J}_{t(2,3),1}\mathbf{J}_{t2,2}^\top & \cdots & \mathbf{J}_{t(k,k+1),1}\mathbf{J}_{tk,k}^\top \\ \mathbf{0} & \mathbf{J}_{t(2,3),2}\mathbf{J}_{t2,2}^\top - \mathbf{G}_{t2,22} & \cdots & \mathbf{J}_{t(k,k+1),2}\mathbf{J}_{tk,k}^\top \\ \vdots & \vdots & \ddots & \vdots \\ \mathbf{0} & \mathbf{0} & \cdots & \mathbf{J}_{t(k,k+1),k}\mathbf{J}_{tk,k}^\top - \mathbf{G}_{tk,kk} \end{bmatrix}.$$

*Proof.* (a) For (38), this is exactly the update formula (10) for the $i$-th batch where the Jacobian and Gram matrix are $\mathbf{J}_{ti,i}$ and $\mathbf{G}_{ti,ii}$ respectively. Note that

$$\mathbf{G}_{ti,ii}^{-1}(\mathbf{f}_{ti,i} - \mathbf{y}_i) = \begin{bmatrix} \mathbf{0}_{b \times (i-1)b} & \mathbf{G}_{ti,ii}^{-1} & \mathbf{0}_{b \times (k-i)b} \end{bmatrix}(\mathbf{f}_{ti} - \mathbf{y}),$$

we then obtain (39) from (38).

(b). We have

$$\begin{aligned} &(\mathbf{f}_{t(i+1)} - \mathbf{y}) - (\mathbf{f}_{ti} - \mathbf{y}) \\ &= \mathbf{f}_{t(i+1)} - \mathbf{f}_{ti} \\ &= \mathbf{J}_{t(i,i+1)}(\mathrm{vec}(\mathbf{W}_{t(i+1)}) - \mathrm{vec}(\mathbf{W}_{ti})) \\ &= \mathbf{J}_{t(i,i+1)}\mathbf{J}_{ti,i}^\top\tilde{\mathbf{G}}_{ti}(\mathbf{f}_{ti} - \mathbf{y}) \qquad \text{(By formula (39))}, \end{aligned}$$

we obtain (40).

(c). Based on (40) we know that

$$\mathbf{f}_{t(i+1)} - \mathbf{y} = \prod_{i'=i}^{1}(\mathbf{I}_n - \mathbf{J}_{t(i',i'+1)}\mathbf{J}_{ti',i'}^\top\tilde{\mathbf{G}}_{ti'})(\mathbf{f}_{ti} - \mathbf{y}),$$

where the index goes in decreasing order from left to right. So in order to prove (41) we only need to prove that

$$\left[\prod_{i'=i}^{1}(\mathbf{I}_n - \mathbf{J}_{t(i',i'+1)}\mathbf{J}_{ti',i'}^\top\tilde{\mathbf{G}}_{ti'})\right](\mathbf{D}_t - \mathbf{L}_t) = \mathbf{U}_{ti}, \tag{44}$$

which we will prove by induction on $i$. For $i = 0$ it is trivial that $\mathbf{D}_t - \mathbf{L}_t = \mathbf{U}_{t0}$ by definition. Suppose (44) holds for $i$, then

$$\begin{aligned} &\left[\prod_{i'=i+1}^{1}(\mathbf{I}_n - \mathbf{J}_{t(i',i'+1)}\mathbf{J}_{ti',i'}^\top\tilde{\mathbf{G}}_{ti'})\right](\mathbf{D}_t - \mathbf{L}_t) \\ &= (\mathbf{I}_n - \mathbf{J}_{t(i+1,i+2)}\mathbf{J}_{t(i+1),i+1}^\top\tilde{\mathbf{G}}_{t(i+1)})\mathbf{U}_{ti} \\ &= \mathbf{U}_{ti} - \mathbf{J}_{t(i+1,i+2)}\mathbf{J}_{t(i+1),i+1}^\top\mathbf{G}_{t(i+1),(i+1)(i+1)}^{-1}\mathbf{U}_{ti}((ib+1:(i+1)b, 1:n)) \\ &= \mathbf{U}_{ti} - \begin{bmatrix} \mathbf{J}_{t(i+1,i+2),1}\mathbf{J}_{t(i+1),i+1}^\top \\ \mathbf{J}_{t(i+1,i+2),2}\mathbf{J}_{t(i+1),i+1}^\top \\ \vdots \\ \mathbf{J}_{t(i+1,i+2),k}\mathbf{J}_{t(i+1),i+1}^\top \end{bmatrix}\begin{bmatrix} \mathbf{0}_{b \times (ib)} & \mathbf{I}_{b \times b} & \mathbf{0}_{b \times (k-i-2)b} \end{bmatrix} \\ &= \mathbf{U}_{t(i+1)}, \end{aligned}$$

which proves (41). Note that by the definition, we have $\overline{\mathbf{U}}_{ti} = \overline{\mathbf{D}}_{ti} - \overline{\mathbf{L}}_{ti}$, we can then obtain (42) by

$$\begin{aligned} \mathbf{U}_{ti}(\mathbf{D}_t - \mathbf{L}_t)^{-1} &= \begin{bmatrix} \tilde{\mathbf{U}}_{ti} & \mathbf{0} \\ \mathbf{0} & \overline{\mathbf{U}}_{ti} \end{bmatrix}\begin{bmatrix} \tilde{\mathbf{D}}_{ti} - \tilde{\mathbf{L}}_{ti} & \mathbf{0} \\ \hat{\mathbf{D}}_{ti} - \hat{\mathbf{L}}_{ti} & \overline{\mathbf{D}}_{ti} - \overline{\mathbf{L}}_{ti} \end{bmatrix}^{-1} \\ &= \begin{bmatrix} \tilde{\mathbf{U}}_{ti} & \mathbf{0} \\ \mathbf{0} & \overline{\mathbf{U}}_{ti} \end{bmatrix}\begin{bmatrix} (\tilde{\mathbf{D}}_{ti} - \tilde{\mathbf{L}}_{ti})^{-1} & \mathbf{0} \\ -(\overline{\mathbf{D}}_{ti} - \overline{\mathbf{L}}_{ti})^{-1}(\hat{\mathbf{D}}_{ti} - \hat{\mathbf{L}}_{ti})(\tilde{\mathbf{D}}_{ti} - \tilde{\mathbf{L}}_{ti})^{-1} & (\overline{\mathbf{D}}_{ti} - \overline{\mathbf{L}}_{ti})^{-1} \end{bmatrix} \\ &= \begin{bmatrix} \tilde{\mathbf{U}}_{ti}(\tilde{\mathbf{D}}_{ti} - \tilde{\mathbf{L}}_{ti})^{-1} & \mathbf{0} \\ -(\hat{\mathbf{D}}_{ti} - \hat{\mathbf{L}}_{ti})(\tilde{\mathbf{D}}_{ti} - \tilde{\mathbf{L}}_{ti})^{-1} & \mathbf{I} \end{bmatrix}. \end{aligned}$$

$\square$

By (43), we can see that the iteration matrix $\mathbf{A}_t$ is close to the matrix $\mathbf{A}$ defined in (15). The convergence of the algorithm is much related to the eigenvalues of $\mathbf{A}$. In the next two lemmas, we bound the spectral radius of $\mathbf{A}$ and provide an auxiliary result on the convergence on perturbed matrices based on the spectral radius.

**Lemma 7.** *Suppose the least eigenvalue of the initial Gram matrix $\mathbf{G}_0$ satisfies $\lambda_{\min}(\mathbf{G}_0) \geq \frac{3}{4}\lambda_0$, (which is true with probability $1 - \delta$ if $M = \Omega\left(\frac{n^2 \log(2n/\delta)}{\lambda_0^2}\right)$, according to Lemma 3). Also assume $\|J_{\mathbf{W}_0, \mathbf{x}_l}\|_F = O(1)$ for any $l \in [n]$ (which is true with probability $1 - \delta$, according to Lemma 2). Then the spectral radius $\mathbf{A}$, or equivalently, maximum norm of eigenvalues of $\mathbf{A}$, denoted $\rho(\mathbf{A})$, satisfies*

$$\rho(\mathbf{A}) \leq 1 - \Omega\left(\frac{\lambda_0^2}{n^2}\right). \tag{45}$$

*Proof.* For any eigenvalue $\lambda \in \mathbb{C}$ of $\mathbf{A}$, it is an eigenvalue of $\mathbf{A}^\top$, so there exists a corresponding unit vector $\mathbf{v} \in \mathbb{C}^{n \times 1}$ such that $\lambda \mathbf{v} = \mathbf{A}^\top \mathbf{v} = (\mathbf{D} - \mathbf{L}^\top)^{-1}\mathbf{L}\mathbf{v}$, which means

$$\lambda(\mathbf{D} - \mathbf{L}^\top)\mathbf{v} - \mathbf{L}\mathbf{v} = 0. \tag{46}$$

Since $\mathbf{G}_0 = \mathbf{D} - \mathbf{L} - \mathbf{L}^\top$ is positive definite with eigenvalues at least $\frac{3}{4}\lambda_0$, we have $\mathbf{v}^*(\mathbf{D} - \mathbf{L} - \mathbf{L}^\top)\mathbf{v} \geq \frac{3}{4}\lambda_0$, which means

$$d - 2\operatorname{Re}(l) \geq \frac{3}{4}\lambda_0, \tag{47}$$

where $d = \mathbf{v}^*\mathbf{D}\mathbf{v}$, $l = \mathbf{v}^*\mathbf{L}\mathbf{v}$, and $\mathbf{v}^*\mathbf{L}^\top\mathbf{v} = \bar{l}$. Let $\mathbf{v}_i = \mathbf{v}((i-1)b + 1 : ib) \in \mathbb{R}^{b \times 1}$ be the $i$-th batch of the vector $\mathbf{v}$, and $\tilde{\mathbf{v}}_i = [\mathbf{0}_{1 \times (i-1)b}, \mathbf{v}_i^\top, \mathbf{0}_{1 \times (k-i)b}]^\top \in \mathbb{R}^{n \times 1}$. It is not hard to see that

$$d = \mathbf{v}^*\mathbf{D}\mathbf{v} = \sum_{i=1}^{k}\mathbf{v}_i^*\mathbf{G}_{0,ii}\mathbf{v}_i = \sum_{i=1}^{k}\tilde{\mathbf{v}}_i^*\mathbf{G}_0\tilde{\mathbf{v}}_i \geq \sum_{i=1}^{k}\frac{3}{4}\lambda_0\|\tilde{\mathbf{v}}_i\|_2^2 = \frac{3}{4}\lambda_0. \tag{48}$$

Also, since by our assumption each entry of $\mathbf{L}$ (or of $\mathbf{G}_0$) is at most $O(1)$, we have

$$|l| = \mathbf{v}^*\mathbf{L}\mathbf{v} \leq \|\mathbf{L}\|_2 \leq \|\mathbf{L}\|_F = O(n). \tag{49}$$

Now we use take an inner product of $\mathbf{v}$ with (46) and get

$$0 = \mathbf{v}^*(\lambda(\mathbf{D}\mathbf{v} - \mathbf{L}^\top\mathbf{v}) - \mathbf{L}\mathbf{v}) = \lambda(d - \bar{l}) + l, \tag{50}$$

and therefore

$$
\begin{aligned}
|\lambda| &= \left|\frac{l}{d - \bar{l}}\right| \qquad \text{(By solving (50))} \\
&= \sqrt{\frac{|l|^2}{(d - l)(d - \bar{l})}} = \sqrt{\frac{1}{\frac{d(d - 2\operatorname{Re}(l))}{|l|^2} + 1}} \\
&\leq \sqrt{\frac{1}{\Omega\left(\frac{\lambda_0^2}{n^2}\right) + 1}} \qquad \text{(Using (47), (48), (49))} \\
&= 1 - \Omega\left(\frac{\lambda_0^2}{n^2}\right).
\end{aligned}
$$

This concludes the proof for this lemma. $\qquad\square$

**Lemma 8.** *Denote $\rho(\mathbf{A}) = \rho_0 \leq 1$. Let $\mathbf{A}$ be diagonalized as $\mathbf{A} = \mathbf{P}^{-1}\mathbf{Q}\mathbf{P}$ and $\mu = \|\mathbf{P}\|_2\|\mathbf{P}^{-1}\|_2$ (see Assumption 2). Suppose we have $\|\mathbf{A}_t - \mathbf{A}_0\|_2 \leq \delta$ for $t \leq T$, then*

$$\left\|\prod_{i=1}^{T}\mathbf{A}_t\right\|_2 \leq \mu(\rho + \mu\delta)^T$$

*Proof.* We have

$$
\left\| \prod_{i=1}^{T} \mathbf{A}_t \right\|_2 = \left\| \prod_{i=1}^{T} (\mathbf{A} + (\mathbf{A}_t - \mathbf{A}_0)) \right\|_2
$$

$$
= \left\| \prod_{i=1}^{T} \mathbf{P}^{-1} \left( \mathbf{Q} + \mathbf{P}(\mathbf{A}_t - \mathbf{A}_0)\mathbf{P}^{-1} \right) \mathbf{P} \right\|_2
$$

$$
= \left\| \mathbf{P} \left( \prod_{i=1}^{T} \left( \mathbf{Q} + \mathbf{P}(\mathbf{A}_t - \mathbf{A}_0)\mathbf{P}^{-1} \right) \right) \mathbf{P} \right\|_2
$$

$$
\leq \|\mathbf{P}\|_2 \left( \prod_{t=1}^{T} \left( \|\mathbf{Q}\|_2 + \|\mathbf{P}\|_2 \|\mathbf{A}_t - \mathbf{A}_0\|_2 \|\mathbf{P}\|_2 \right) \right) \|\mathbf{P}^{-1}\|_2
$$

$$
\leq \mu(\rho + \mu\delta)^T.
$$

$\square$

In addition to the bounds used in Theorem 1, we provide the next lemma with useful bounds of the norms and eigenvalues of relevant matrices in the optimization scope

$$
B(R) = \{\mathbf{W} : \|\mathbf{W} - \mathbf{W}_0\|_F \leq R\}.
$$

**Lemma 9** (Relevant Bounds for the Matrices in the Optimization Scope). *We assume the events in Lemma 2 and 3 hold, and let $M = \frac{C\mu^2 n^8 R^2}{\lambda_0^8}$ for some large enough constant $C$, then:*

*For any $(t, i)$-pair, we assume $\mathbf{W}_{t'i'} \in B(R)$ for all $i' \in [k]$ when $t' \leq t$ and $i' \in [i]$ when $t' = t$ in the following propositions.*

*(a). $\min_{\mathbf{v} \in \mathbb{R}^n, \|\mathbf{v}\|_2 = 1} \|(\mathbf{D} - \mathbf{L})\mathbf{v}\|_2 \geq \frac{3}{8}\lambda_0$.*

*(b). Suppose up to $t$, $\mathbf{W}_t$ is in $B(R)$ (which means for $i' \in [k+1]$, and $t' \in [t-1]$, $W_{ti} \in B(R)$). Then $\|\mathbf{A}_t - \mathbf{A}\|_2 \leq \frac{1-\rho(\mathbf{A})}{2\mu}$.*

*Proof.* (a). Because for $\|\mathbf{v}\|_2 = 1$,

$$
\|(\mathbf{D} - \mathbf{L})\mathbf{v}\|_2 \geq \mathbf{v}^\top (\mathbf{D} - \mathbf{L})\mathbf{v}
$$

$$
= \frac{1}{2}\mathbf{v}^\top \mathbf{D}\mathbf{v} + \frac{1}{2}\mathbf{v}^\top (\mathbf{D} - \mathbf{L} - \mathbf{L}^\top)\mathbf{v}
$$

$$
\geq 0 + \frac{1}{2}\lambda_{\min}(\mathbf{G}_0)
$$

$$
\geq \frac{3}{8}\lambda_0. \qquad \text{(By the positive-definiteness of } \mathbf{G}_0 \text{ and Lemma 3)}
$$

(b). By Lemma 4 we know that within $B(R)$, the size of the Jacobian $\|J_{\mathbf{x}_l}\|_F$ w.r.t. data $\mathbf{x}_l$ is $O(1)$, and the difference $\|J_{\mathbf{W}_1, \mathbf{x}_l} - J_{\mathbf{W}_2, \mathbf{x}_l}\|_F$ within $O(R)$ is bounded by $O\left(\frac{R}{\sqrt{M}}\right)$, this can be applied to each entry of $\mathbf{D}, \mathbf{L}, \mathbf{U}$, etc., including those $\mathbf{J}_{t'(i', i'+1)}$ terms by the convexity of $B(R)$, and we can see that each entry of these matrices has size at most $O(1)$ and varies inside an $O\left(\frac{R}{\sqrt{M}}\right)$ range. Therefore we get

$$
\|\mathbf{U}_t\|_F = O(n),
$$
$$
\|\mathbf{D}_t - \mathbf{L}_t\|_F = O(n),
$$

and

$$
\|\mathbf{U}_t - \mathbf{U}\|_2 = O\left(\frac{Rn}{\sqrt{M}}\right),
$$

$$
\|(\mathbf{D}_t - \mathbf{L}_t) - (\mathbf{D} - \mathbf{L})\|_2 = O\left(\frac{Rn}{\sqrt{M}}\right).
$$

By our chosen $M$, $O\left(\frac{Rn}{\sqrt{M}}\right) \le \frac{1}{8}\lambda_0$ can definitely hold, so along with (a) we have

$$\left\|(\mathbf{D}-\mathbf{L})^{-1}\right\|_2 = O\left(\frac{1}{\lambda_0}\right), \left\|(\mathbf{D}_t-\mathbf{L}_t)^{-1}\right\|_2 = O\left(\frac{1}{\lambda_0}\right).$$

Hence,

$$
\begin{aligned}
\|\mathbf{A}_t - \mathbf{A}\|_2 &= \left\|(\mathbf{U}_t(\mathbf{D}_t-\mathbf{L}_t)^{-1} - \mathbf{U}(\mathbf{D}-\mathbf{L})^{-1}\right\|_2 \\
&= \left\|(\mathbf{U}_t - \mathbf{U})(\mathbf{D}_t-\mathbf{L}_t)^{-1} + \mathbf{U}((\mathbf{D}_t-\mathbf{L}_t)^{-1}((\mathbf{D}_t-\mathbf{L}_t)-(\mathbf{D}-\mathbf{L}))(\mathbf{D}-\mathbf{L})^{-1}\right\|_2 \\
&\le O\left(\frac{Rn}{\sqrt{M}}\right)O\left(\frac{1}{\lambda_0}\right) + O(n)O\left(\frac{1}{\lambda_0}\right)O\left(\frac{Rn}{\sqrt{M}}\right)O\left(\frac{1}{\lambda_0}\right) \\
&= O\left(\frac{n^2 R}{\lambda_0^2 \sqrt{M}}\right),
\end{aligned}
$$

and along with Lemma 7, in order to have $\|\mathbf{A}_t - \mathbf{A}\|_2 \le \frac{1-\rho(\mathbf{A})}{2\mu}$, all we need is

$$O\left(\frac{n^2 R}{\lambda_0^2 \sqrt{M}}\right) \le \Omega\left(\frac{\lambda_0^2}{n^2}\right),$$

so choosing some $M = \frac{C\mu^2 n^8 R^2}{\lambda_0^8}$ suffices. $\qquad\square$

With all the preparations, now we are ready to prove Theorem 2. The logic of the proof is just the same as what we did in the formal proof of Theorem 1, where we then used induction on $t$ and now we use induction on the pair $(t, i)$. The key, is still selecting some $R$ so that in each step $\mathbf{W}_{ti}$ remains in $B(R)$. Combined with the previous Lemmas, in $B(R)$ we have $\mathbf{A}_t$ being close to $\mathbf{A}$, and then convergence is guaranteed.

**Formal proof of Theorem 2.** Let $R_{ti} = \left\|\mathbf{W}_{t(i+1)} - \mathbf{W}_{ti}\right\|_F$. We take

$$R = \Theta\left(\frac{n^5}{\lambda_0^4}\right)$$

in the range $B(R)$ (where the constant is chosen to make the right hand side of (52) hold). We prove that there exists an

$$M = \max\left(\Omega\left(\frac{\mu^2 n^{18}}{\lambda_0^{16}}\right), \Omega\left(\frac{n^2 d \log(16n/\delta)}{\lambda_0^2}\right)\right)$$

with a large enough constant that suffices. First we can easily verify that all the requirements for $M$ in Lemma 2-9 (most importantly, Lemma 9) can be satisfied. Hence, with probability at least $1-\delta$ all the events in Lemma 2-9 hold. Under this situation, we do induction on $(t,i)$ ($t \in \{1,2,\cdots\}, i \in [k]$, by dictionary order) to show that:

- $\mathbf{W}_{ti} \in B(R)$.

For $(t, i) = (1, 1)$, it holds by definition.

Suppose the proposition holds up to $(t, i)$. Then since $\mathbf{W}_{ti} \in B(R)$, by Lemma 4 we know that $\lambda_{\min}(\mathbf{G}_{ti}) \ge \frac{\lambda_0}{2}$. This naturally gives us $\lambda_{\min}(\mathbf{G}_{ti,ii}) \ge \frac{\lambda_0}{2}$, which means $\mathbf{G}_{ti,ii}$ is invertible.

Similar to the argument in the proof of Lemma 9, we know that each entry of $\mathbf{J}_{ti}, \mathbf{J}_{t(i,i+1)}, \mathbf{D}_{ti}, \mathbf{L}_{ti}, \mathbf{U}_{ti}$, etc., whose index of $(t', i')$ only contains $i'$ with $i' \le i$, is of scale $O(1)$ and varies at most $O\left(\frac{R}{\sqrt{M}}\right)$, which is a result of Lemma 4. Then we know

$$
\begin{aligned}
\|\mathbf{J}_{ti,i}\|_F &= O(\sqrt{b}) = O(\sqrt{n}), \\
\left\|\tilde{\mathbf{D}}_{t(i-1)} - \tilde{\mathbf{L}}_{t(i-1)}\right\|_F &= O(n), \\
\left\|\hat{\mathbf{D}}_{t(i-1)} - \hat{\mathbf{L}}_{t(i-1)}\right\|_F &= O(n),
\end{aligned}
$$

etc. We then also know

$$\left\| (\tilde{\mathbf{D}}_{t(i-1)} - \tilde{\mathbf{L}}_{t(i-1)}) - (\mathbf{D}_t - \mathbf{L}_t)_{(1:(i-1)b,1:(i-1)b)} \right\|_F \leq O\left( \frac{nR}{\sqrt{M}} \right).$$

Since we can also have $\left\| ((\mathbf{D}_t - \mathbf{L}_t) - (\mathbf{D} - \mathbf{L}))_{(1:(i-1)b,1:(i-1)b)} \right\|_F \leq O\left( \frac{nR}{\sqrt{M}} \right)$, and given our choice of $M$ and $R$ the right hand side is less than $\frac{1}{8}\lambda_0$, along with Lemma (9) (which says

$$\min_{\mathbf{v} \in \mathbb{R}^n, \|\mathbf{v}\|_2 = 1} \|(\mathbf{D} - \mathbf{L})\mathbf{v}\|_2 \geq \frac{3}{8}\lambda_0$$

and naturally, since this matrix is block-lower-triangular,

$$\min_{\mathbf{v} \in \mathbb{R}^{(i-1)b}, \|\mathbf{v}\|_2 = 1} \left\| (\mathbf{D} - \mathbf{L})_{1:(i-1)b,1:(i-1)b} \mathbf{v} \right\|_2 \geq \frac{3}{8}\lambda_0$$

holds) we know that $\tilde{\mathbf{D}}_t - \tilde{\mathbf{L}}_t$ is invertible and has the bound $\left\| (\tilde{\mathbf{D}}_t - \tilde{\mathbf{L}}_t)^{-1} \right\|_2 \leq \frac{4}{\lambda_0}$.

Based on the update rule (38), we have

$$
\begin{aligned}
R_{ti} &= \left\| \mathbf{W}_{ti} - \mathbf{W}_{t(i+1)} \right\|_F \\
&= \|\text{vec}(\mathbf{W}_t) - \text{vec}(\mathbf{W}_{t+1})\|_2 \\
&\leq \left\| \mathbf{J}_{ti,i}^\top \mathbf{G}_{ti,ii}^{-1} (\mathbf{f}_{ti,i} - \mathbf{y}_i) \right\|_2 \\
&\leq O\left( \frac{\sqrt{n}}{\lambda_0} \right) \|\mathbf{f}_{ti} - \mathbf{y}\|_2 \qquad\qquad \left( \left\| \mathbf{J}_{ti,i}^\top \right\|_2 = O(\sqrt{n}), \left\| \mathbf{G}_{ti,ii}^{-1} \right\|_2 = O\left( \frac{1}{\lambda_0} \right) \right) \\
&= O\left( \frac{\sqrt{n}}{\lambda_0} \right) \left\| \begin{bmatrix} \tilde{\mathbf{U}}_{t(i-1)}(\tilde{\mathbf{D}}_{t(i-1)} - \tilde{\mathbf{L}}_{t(i-1)})^{-1} & \mathbf{0} \\ -(\hat{\mathbf{D}}_{t(i-1)} - \hat{\mathbf{L}}_{t(i-1)})(\tilde{\mathbf{D}}_{t(i-1)} - \tilde{\mathbf{L}}_{t(i-1)})^{-1} & \mathbf{I}_{(k-i+1)b} \end{bmatrix} \right\|_2 \|\mathbf{f}_t - \mathbf{y}\|_2 \\
&\qquad\qquad\qquad\qquad\qquad\qquad\qquad \text{(By formula (42))} \\
&\leq O\left( \frac{\sqrt{n}}{\lambda_0} \right) \left( \left\| \tilde{\mathbf{U}}_{t(i-1)}(\tilde{\mathbf{D}}_{t(i-1)} - \tilde{\mathbf{L}}_{t(i-1)})^{-1} \right\|_F \right. \\
&\quad + \left\| (\hat{\mathbf{D}}_{t(i-1)} - \hat{\mathbf{L}}_{t(i-1)})(\tilde{\mathbf{D}}_{t(i-1)} - \tilde{\mathbf{L}}_{t(i-1)})^{-1} \right\|_F \\
&\quad + \left. \left\| \mathbf{I}_{(k-i+1)b} \right\|_F \right) \|\mathbf{f}_t - \mathbf{y}\|_2 \\
&\leq O\left( \frac{n^{3/2}}{\lambda_0^2} \right) \|\mathbf{f}_t - \mathbf{y}\|_2 \qquad\qquad\qquad\qquad\qquad\qquad\qquad (51) \\
&\leq O\left( \frac{n^{3/2}}{\lambda_0^2} \right) \left\| \prod_{i=1}^{t-1} \mathbf{A}_t \right\|_2 \|\mathbf{f}_1 - \mathbf{y}\|_2 \qquad \text{(By Eq. (43))} \\
&\leq O\left( \frac{n^{3/2}}{\lambda_0^2} \right) \cdot \mu \left( 1 - \Omega\left( \frac{\lambda_0^2}{n^2} \right) \right)^{(t-1)} \|(\mathbf{f}_1 - \mathbf{y})\|_2 \qquad \text{(Lemma 7, 8, and 9(b).)} \\
&\leq O\left( \frac{\mu n^2}{\lambda_0^2} \right) \left( 1 - \Omega\left( \frac{\lambda_0^2}{n^2} \right) \right)^{(t-1)}, \qquad\qquad\qquad \text{(Lemma 2)}
\end{aligned}
$$

where Eq. (51) uses the following bounds proved above:

$$\left\| \tilde{\mathbf{U}}_{t(i-1)}(\tilde{\mathbf{D}}_{t(i-1)} - \tilde{\mathbf{L}}_{t(i-1)})^{-1} \right\|_F \leq \left\| \tilde{\mathbf{U}}_{t(i-1)} \right\|_F \left\| (\tilde{\mathbf{D}}_{t(i-1)} - \tilde{\mathbf{L}}_{t(i-1)})^{-1} \right\|_2 = O\left( \frac{n}{\lambda_0} \right),$$

$$\left\| (\hat{\mathbf{D}}_{t(i-1)} - \hat{\mathbf{L}}_{t(i-1)})(\tilde{\mathbf{D}}_{t(i-1)} - \tilde{\mathbf{L}}_{t(i-1)})^{-1} \right\|_F \leq \left\| (\hat{\mathbf{D}}_{t(i-1)} - \hat{\mathbf{L}}_{t(i-1)}) \right\|_F \left\| (\tilde{\mathbf{D}}_{t(i-1)} - \tilde{\mathbf{L}}_{t(i-1)})^{-1} \right\|_2 = O\left( \frac{n}{\lambda_0} \right),$$

$$\left\| \mathbf{I}_{(k-i+1)b} \right\|_F = O(n).$$

Since this also hold for previous $(t, i)$ pairs, we have

$$
\sum_{(t', i') \leq (t, i)} R_{t'i'} \leq O\left(\frac{\mu k n^2}{\lambda_0^2}\right) \sum_{t'=1}^{t} \left(1 - \Omega\left(\frac{\lambda_0^2}{n^2}\right)\right)^{(t'-1)}
$$

$$
\leq O\left(\frac{\mu k n^2}{\lambda_0^2}\right) O\left(\frac{n^2}{\lambda_0^2}\right)
$$

$$
\leq O\left(\frac{n^5}{\lambda_0^4}\right) \leq R, \tag{52}
$$

which is the reason why we need to take $R = \Theta\left(\frac{n^5}{\lambda_0^4}\right)$. This means that $\mathbf{W}_{ti} \in B(R)$ holds. And by induction, we have proved that $\mathbf{W}$ remains in $B(R)$ throughout the optimization process.

The last thing to do is to bound $\mathbf{f}_t - \mathbf{y}$. By the same logic from above, we have

$$
\|\mathbf{f}_t - \mathbf{y}\|_2 \leq \left\|\prod_{i=1}^{t-1} \mathbf{A}_t\right\|_2 \|\mathbf{f}_1 - \mathbf{y}\|_2
$$

$$
\leq \mu\sqrt{n}\left(1 - \Omega\left(\frac{\lambda_0^2}{n^2}\right)\right)^{(t-1)},
$$

which proves our theorem.

## D  ADDITIONAL EXPERIMENTAL RESULTS

In this section, we give test performance curve of AFAD-LITE dataset in Figure 3 under the same setting with Section 4.

In addition, we provide more baseline results, e.g. Adam (Kingma & Ba, 2014) and K-FAC (Martens & Grosse, 2015) on RSNA Bone Age dataset. Since we find that, as another alternation of BN, Group Normalization (GN) (Wu & He, 2018) can largely improve the performance of Adam, we also implement the GN layer for our GGN method. We use grid search to obtain approximately best hyper-parameters for every experiment. All experiments are performed with batch size 128, input size 64*64 and weight decay $10^{-4}$. We set the number of groups to 8 for ResNetGN. Other hyper-parameters are listed below.

- SGD+ResNetBN: learning rate 0.01, momentum 0.9.
- Adam+ResNetBN: learning rate 0.001.
- SGD+ResNetGN: learning rate 0.002, momentum 0.9.
- Adam+ResNetGN: learning rate 0.0005.
- K-FAC+ResNet: learning rate 0.02, momentum 0.9, $\epsilon = 0.1$, update frequency 100.
- GGN+ResNetGN: $\lambda = 1$, $\alpha = 0.075$.

The convergence results are summarized in Figure 4. Note to make comparison clearer, we use logarithmic scale for training curves.

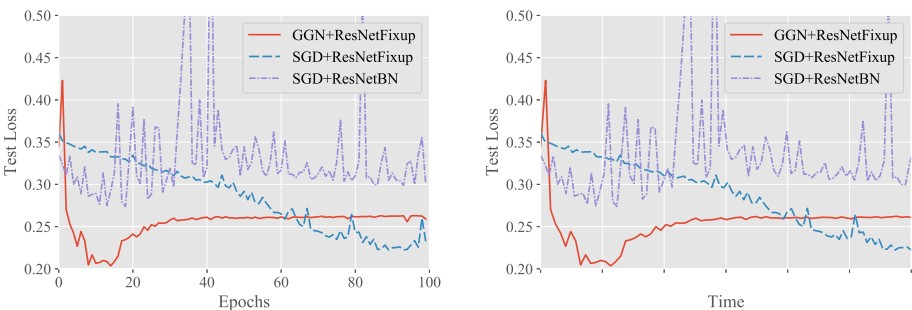

Figure 3: Test performance on AFAD-LITE dataset.

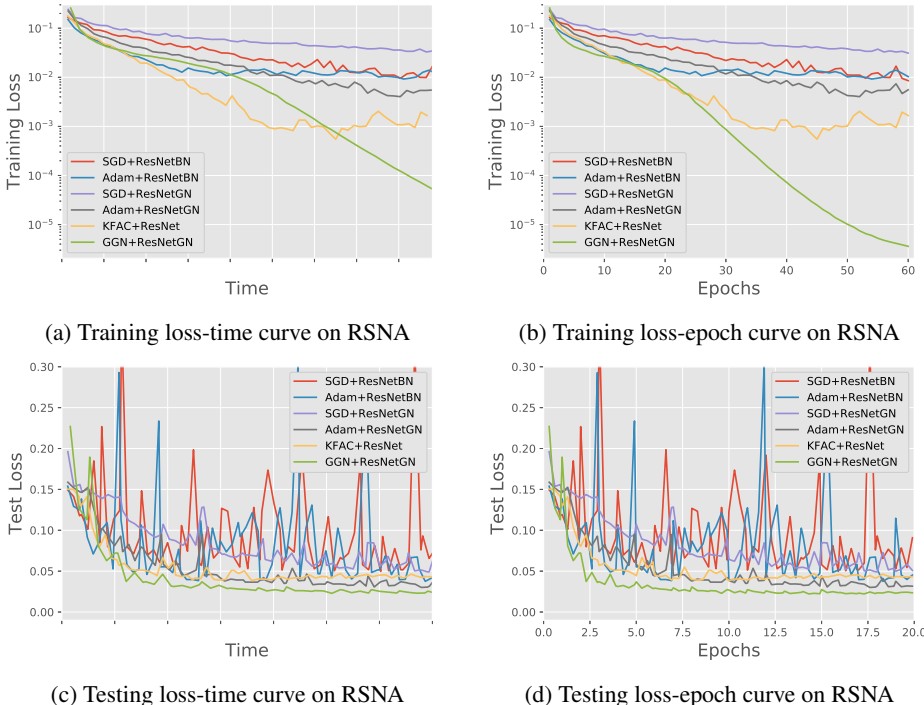

(a) Training loss-time curve on RSNA

(b) Training loss-epoch curve on RSNA

(c) Testing loss-time curve on RSNA

(d) Testing loss-epoch curve on RSNA

Figure 4: Training and testing curves of GGN and other baselines on RSNA Bone Age dataset.

