# OpenReview forum: "Gram-Gauss-Newton Method: Learning Overparameterized Neural Networks for Regression Problems"
_ICLR.cc/2020/Conference — Reject_

### Official Review · AnonReviewer3 · 2019-10-23
**Official Blind Review #3**

**Rating:** 3

**Review:**

Post-rebuttal: I've read author's response and other reviews. As pointed out by other reviewers, the proposed algorithm is restricted to single-output regression and the claim "accelerate convergence without much computational overhead" might not be true in general multi-output regression tasks. I believe the lack of multi-output regression experiments makes the paper a bit weak, therefore I changed my score to 3 and vote for rejection.

That being said, I do find the algorithm interesting and the theoretical results impressive. I encourage the authors to include experiments on multi-output regression tasks (or tone down the claim about computational overhead) and resubmit the paper.

------------------------------------------------------------------------------------------------------------------------------------------------------------------
Based on recent progress on the connection between neural network training and kernel regression of neural tangent kernel, this paper proposes a Gram-Gauss-Newton (GGN) algorithm to train deep neural networks for regression problems with square loss. For overparameterized shallow networks, the authors proved global convergence of the proposed algorithm in both full-batch and mini-batch setting. To my knowledge, the proof of global convergence in the mini-batch setting is novel and might be of independent interest for other work.

Overall, this paper is well-written and easy to follow. It's interesting to see that the proposed algorithm can achieve quadratic convergence while most previous papers only get linear convergence.
Given that, I'd like to give a score of 6 and I'm willing to increase my score if the authors can resolve my concerns below.

Concerns:
- For the algorithm, if I understand correctly, it's actually same as natural gradient descent with generalized inverse. I think the authors should make the connection clear. I would like to see more discussions with natural gradient descent or Newton methods in the next revision.
- The authors claim that the proposed GGN algorithm only has minor computational overhead compared to first-order methods. I doubt if it's true in general. In section 3.3, the authors argue that computing individual Jacobian matrices for every example in the minibatch has roughly the same computation as the backpropagation in SGD. As far as I know, it's not true in practice. In addition, the inverse of the Gram matrix can also be expensive when the output dimension (the dimension of y) is large.

Minor Comments:
- In the paper, the theoretical results are based on the assumption of smooth activation function. I wonder if it is possible to include the case of ReLU activation as it's the most popular activation function in deep learning.
- I don't have a good understanding about why mini-batch version would converge after reading the paper. To me, second-order methods with mini-batch estimation of the preconditioner would lead to biased gradient estimation. Could you comment on that?

**Experience Assessment:**

I have published one or two papers in this area.

**Review Assessment: Checking Correctness Of Derivations And Theory:**

I assessed the sensibility of the derivations and theory.

**Review Assessment: Checking Correctness Of Experiments:**

I assessed the sensibility of the experiments.

**Review Assessment: Thoroughness In Paper Reading:**

I read the paper at least twice and used my best judgement in assessing the paper.

---

> ### Author Response · Authors · 2019-11-15
> **Response**
>
> Thank you for your valuable comments. We have addressed the most common issues in the general response above. Here we answer the additional questions raised by the reviewer.
>
> --Activation functions. We believe it is possible to change our proof to ReLU activations based on the techniques in [1].
>
> --Proof techniques of mini-batch GGN. As mentioned in Section 1, though conventional wisdom may suggest that applying mini-batch scheme to second-order methods will introduce a biased estimation of the accelerated gradient direction, we can prove that mini-batch GGN converges on overparametrized networks. Our proof only entails the decrease of the loss after performing a whole cycle of updates. This is significantly different from the former techniques used to prove the convergence of SGD, which uses a small learning rate to force the decrease of expected loss at each step.
>
> [1] Gradient descent provably optimizes over-parameterized neural networks, Du et al.

---

### Official Review · AnonReviewer2 · 2019-10-27
**Official Blind Review #2**

**Rating:** 3

**Review:**

The paper presents a second order optimization algorithm, along with convergence proof of the algorithm in both batch and minibatch setting. The effectiveness of the method is demonstrated on two regression tasks. My overall assessment is that the method is still quite limited and the method itself is not novel, but I am willing to change my score to accept if my concerns have been addressed.

(1) The method is not novel. The same algorithm was proposed and applied to the RL setting [1].
(2) The method is still quite limited to 1-output function scenario, where the NTK matrix is easy to compute. This limitation though is not mentioned in the paper. I hope the author should have a discussion on this and admit this limitation.
(3) Also due to (2), the experiments shown in the paper are on toy data and hence lack of strong empirical support.
(4) The method doesn't scale up to large batch size.
(5) In the theoretical section, the paper states
"However, to our knowledge, no convergence result considering large learning rate (e.g. has the same scale with the
update of GGN) has been proposed."
This is not true. Here are some papers: [2,3,4]
(6) Lack of some second order optimization baselines, e.g., KFAC.

Misc:
(1) For section 3.3, first of all, (B) costs at least half of (A) as it requires a backward pass.
(2) For section 3.3, the authors write:
" What is different is that GGN also, for every input data, keeps track of the output’s derivative for the parameters; while in
SGD the derivatives for the parameters are averaged over a batch of data."
Is there a simple way of implementing/computing the gradient for *every* input data on GPU? How is that compared to computing the average? I wish to see more evidence of showing they're the same as authors claimed.

[1] Towards Characterizing Divergence in Deep Q-Learning.
[2] The Power of Interpolation: Understanding the Effectiveness of SGD in Modern Over-parametrized Learning.
[3] Fast and Faster Convergence of SGD for Over-Parameterized Models (and an Accelerated Perceptron).
[4] Fast Convergence of Stochastic Gradient Descent under a Strong Growth Condition

**Experience Assessment:**

I have published in this field for several years.

**Review Assessment: Checking Correctness Of Derivations And Theory:**

I assessed the sensibility of the derivations and theory.

**Review Assessment: Checking Correctness Of Experiments:**

I assessed the sensibility of the experiments.

**Review Assessment: Thoroughness In Paper Reading:**

I read the paper thoroughly.

---

> ### Author Response · Authors · 2019-11-15
> **Response**
>
> Thank you for your valuable comments. We have addressed the most common issues in the general response above. Here we answer the additional questions raised by the reviewer.
>
> --The RL paper. Thanks for a good reference. The independent work on reinforcement learning aims to precondition the Q-learning update rule with linear approximation, so similar to natural gradient analyzed in [1] , there is still a learning rate term $\alpha$ in the algorithm. However, our method is motivated by solving NTK regression, which does not introduce the step size term (or can be understood as suggesting the learning rate to be 1 as mentioned in the related work section). We added the reference to related work section in the revision.
>
> --Convergence result considering a large learning rate. Thanks for pointing out the misleading expression on large learning rate. As a second-order method, GGN does a Newton-type update without learning rate. Thus, unlike the papers mentioned by the reviewer which need to bound the learning rate by a quantity related to the smoothness to ensure a similar behavior as gradient descent, we show that mini-batch GGN can converge without forcing a specific small step size, which is totally different from the convergence of gradient descent. We modified the expression in the revision.
>
> [1] Fast convergence of natural gradient descent for overparameterized neural networks, Zhang et al.

---

### Official Review · AnonReviewer1 · 2019-10-28
**Official Blind Review #1**

**Rating:** 1

**Review:**

Authors propose minimizing neural network using kernel ridge regression. (Formula 9 and Algorithm 1). Main difference of this method is compared to Gauss-Newton, is that it uses JJ' as curvature, which has dimensions b-by-by (batch size b), instead of J'J as curvature, which has dimensions m-by-m (number of parameters m).

When b is much smaller than m, this matrix is tractable to represent exactly. Related approach is taken by KKT (see Figure 1 of https://arxiv.org/pdf/1806.02958.pdf) which also replaces J'J with more tractable JJ'.

There is a long history of authors trying to extend second order methods to deep learning and and finding that curvature estimated on a small batch is extremely noisy, requiring large batches (see papers by Nocedal's group). Authors propose a method that estimates curvature from small batches. Given the history of failures in small-batch curvature estimation, the bar is high to show that small-batch curvature estimation works.

Bulk of the paper is dedicated to theoretical convergence and connections between concepts. Since the focus of the paper is on a new optimization method for deep learnning, I feel like convergence proofs can be moved to Appendix, and more of the paper should focus on practical aspects of the method. Also the connections to other concepts (ie, tangent kernel) are not essential to the paper and could be better left over for a tutorial paper.

I'm not convinced that their method works well enough to have practical impact.

- Their method seems to be limited to neural network with one output (ie, univariate regression task). This is a serious limitation and paper should highlight this more on this, given that vast majority of applications and benchmarks involve more than output variable.

- Practical implementation details are skimmed over. Section 3.3 brings up that to compute Jacobian, one needs to keep track of the output derivative on per-example basis. How is this accomplished? Modern frameworks like PyTorch and TensorFlow don't give an easy way to compute per-example derivatives efficiently.

- Experiments are performed on two tasks that are not well known in the literature. The choice is somewhat understandable given that their method performs for univariate regression, but also this makes it hard to evaluate whether the method works. SGD vs Gram-Gauss evaluation use parameter settings which are not comparable, so it's impossible to tell whether the improvement are due to better choice of hyper-parameters.


The changes needed to make this paper acceptable are extensive, and I would recommed a reject.

I would recommend authors attempt the following changes for future submission:

1. Make it work for multivariate regression. There's a conversion technique to represent multivariate regression in the same form as univariate regression (see Section 2.4 of "Rao, Toutenberg" Linear Models). Essentially it comes down concatenating o output Jacobians (o output classes) along the batch dimension.

2. Use this to evaluate the method on standard benchmarks like MNIST and CIFAR and show that it doesn't cause a significant worsening in quality. Given that similar approach (KKT paper) found bigger improvement on RNN task, an RNN task may be useful.

3. Give more details on implementation. How was Jacobian calculation implemented? Which framework? How was the per-example computation made tractable? Making small-scale experiments reproducible through anonymous github submission would also help

**Experience Assessment:**

I have published one or two papers in this area.

**Review Assessment: Checking Correctness Of Derivations And Theory:**

I assessed the sensibility of the derivations and theory.

**Review Assessment: Checking Correctness Of Experiments:**

I carefully checked the experiments.

**Review Assessment: Thoroughness In Paper Reading:**

I read the paper thoroughly.

---

> ### Author Response · Authors · 2019-11-15
> **Response**
>
> Thank you for your valuable comments. Most of the issues are addressed in the general response above.

---

### Official Review · AnonReviewer4 · 2019-11-02
**Official Blind Review #4**

**Rating:** 3

**Review:**

The authors propose a scalable second order method for optimization using a quadratic loss. The method is inspired by the Neural Tangent kernel approach, which also allows them to provide global convergence rates for GD and batch SGD. The algorithm has a computational complexity that is linear in the number of parameters and requires to solve a system of the size of the minibatch.  They also show experimentally the advantage of using their proposed methods over SGD.

	- The paper is generally easy to read except section 3.1 which could be clearer when establishing the connexion between the proposed algorithm and NTK.

	- The proposed algorithm seems to be literally a regularized Gauss-Newton with Woodbury matrix inversion lemma applied to equation (7). Additional simplifications occur due to the pre-multiplication by the jacobian and give (9). However, this is not clear in the paper, instead section 3.1, is a bit vague about the derivation of (9).
	- In terms of theory, the proofs of thm 1 and 2 seem sound. They rely essentially on the convergence results established for NTK in [Jacot2018, Chizat2018]. The main novelty is that the authors provide faster rates for the Gauss-Newton pre-conditioner which leads to second-order convergence. The second theoretical contribution is to extend the proof to batched gradient descent. Both are somehow expected, although the second one is more technical.
	- However, the convergence rates provided for batched gradient descent (thm 2) rely on a rather unrealistic assumption: the size of the network should grow as n^18 where n is the sample size. This makes the result less appealing as in practice this is highly unlikely to be the case.
	-  The convergence analysis for the NTK dynamics, which is essential in the proof, relies on a particular scaling 1/sqrt(M) of the function with the number of parameters. In [Chizat2018], it is discussed that although it leads to convergence in the training loss, generalization can be bad. Is there any reason to think in this case, things would be different?

	- Experiments: Experiments were done on two datasets to solve a regression task. They show that training loss decreases indeed faster than SGD and finds better solutions. A more fair comparison would be against other second-order optimizers like KFAC.
	- How was the learning rate chosen for the other methods? Was the same lr used?
	- The authors say that the algorithm has the same cost of one backward pass, could they be more specific about the implementation?
	- What are the test results for the second dataset? Could they be reported somewhere (in the appendix?)
        - Both tasks are univariate regression, can the method be applied successfully in a multivariate setting?
I don't see how the proposed method is different from exactly doing regularized gauss newton, so to me the algorithm is not novel in itself. Besides the method seems to require a quadratic loss function which limits its application.


----------------------------------------------------------------------------------
Revision:


 I've read the author's response and other reviews. I think the paper will be stronger if extended to more general cases (multivariate output + more general losses), thus I encourage the authors to resubmit the paper with stronger experiments.






**Experience Assessment:**

I have published one or two papers in this area.

**Review Assessment: Checking Correctness Of Derivations And Theory:**

I assessed the sensibility of the derivations and theory.

**Review Assessment: Checking Correctness Of Experiments:**

I carefully checked the experiments.

**Review Assessment: Thoroughness In Paper Reading:**

I read the paper thoroughly.

---

> ### Author Response · Authors · 2019-11-15
> **Response**
>
> Thank you for your valuable comments. Most of the issues are addressed in the general response above.

---

### Official Review · AnonReviewer5 · 2019-11-03
**Official Blind Review #5**

**Rating:** 6

**Review:**

Summary: The authors propose the Gram-Gauss-Newton method for training neural networks. Their method draws inspiration from the connection between the neural network optimization and kernel regression of neural tangent kernel.

Their method is described in Algorithm 1, but to summarize it, they use the Gauss-Newton method to train neural networks, and prove quadratic convergence for the full-batch training. They also have a mini-batch version of GGN, the practical version, and this is proven to have linear convergence.

The authors also provide experiments that includes the usual loss v. epoch, but also loss v. wallclock time (which is nice when proposing second-order-like methods where extra computations are necessary), and a test error v. epoch (which is again nice for second-order methods as explained below).

Strengths: The paper has nice proofs of the theorems, and they show a method with quadratic convergence (but full-batch training) without having to invert the full Jacobian matrix whose size depends on the number of parameters, but rather inverting the Gram matrix, whose size depends on the number of training data.

Due to the seeming extra computational cost of the method, (the method requires computing the full Jacobian matrix which depends on the number of neural network weights) I am grateful that they provided comparisons with wallclock time to SGD.

And there is this notion that second-order methods have been shown to not generalize as well as first-order methods, and thus it was nice to see that they had an experiment where they tested generalization.

The background information was also nice to read.

Weaknesses: They do not compare it with other methods optimization methods, such as Adam (a first-order method) or natural gradient (a second-order method), and I would have thus liked to have seen comparisons to these.

I would have also liked to see a test loss v. time/epoch for the AFAD-LITE task as well (they only have it for the RSNA Bone Age task), at least provided in the appendix if there was not enough space.

In the references, there are numerous citations of the arXiv versions of papers, but I suggest the authors replace them with the conference/journal versions if those papers were accepted in conferences/journals (and I spot some that were).

Other comments: (i) In the first sentence of 3.3., I think one should replace “GGN has quadratic convergence rate” with “full-batch GGN has quadratic convergence rate,” as in the subsequent sections you are discussing mini-batch GGN.

**Experience Assessment:**

I have read many papers in this area.

**Review Assessment: Checking Correctness Of Derivations And Theory:**

I assessed the sensibility of the derivations and theory.

**Review Assessment: Checking Correctness Of Experiments:**

I assessed the sensibility of the experiments.

**Review Assessment: Thoroughness In Paper Reading:**

I read the paper at least twice and used my best judgement in assessing the paper.

---

> ### Author Response · Authors · 2019-11-15
> **Response**
>
> Thank you for your valuable comments. Most of the issues are addressed in the general response above.

---

### Author Response · Authors · 2019-11-15
**Response to all reviewers**

We thank all the reviewers for the valuable comments. We address the major issues here which are mentioned multiple times by the reviewers.

--Novelty of the algorithm. The equivalence of natural gradient descent and Gauss-Newton algorithm has been well studied. However, exact solving wasn’t tractable before and people had to rely on approximation methods like K-FAC. We believe that the novelty of GGN lies in the specific implementation of an exact solution by the Gram matrix over a mini-batch scheme, which can be practically useful.

--Limitations of the algorithm. We acknowledge that the current GGN algorithm is still limited to the single-output regression problem. Then again, regression is a fundamental problem in machine learning and already has a large number of application scenarios. As for classification and multivariable regression tasks, as discussed in section 5, the direct application of GGN requires a linear scaling of the size of Jacobian w.r.t. the number of classes. There are possible ways to address this issue, like making some modifications of the network output. This is an important future work, and we are already doing experiments on classification tasks like CIFAR and Imagenet.

--Computational complexity and implementation. Though modern frameworks like PyTorch and TensorFlow don't give an easy way to compute per-example derivatives efficiently, we re-implement the backpropagation process for different type of layers, e.g. convolutional layers, linear layers which makes the computation of Jacobian efficient. We note that a concurrent work [https://openreview.net/forum?id=BJlrF24twB ](Sec 2.2) gives some examples of efficient implementation. We’re still working on making the code cleaner and will release the code if the paper is accepted.

--Experimental results. As requested by the reviewers, we have added the comparison with Adam and K-FAC, as well as the generalization result of AFAD-LITE, in Appendix D.

In general, our paper aims to propose an algorithm that makes use of second-order information to accelerate convergence without much computational overhead, and both the theoretical and experimental results demonstrate its effectiveness. We agree with the reviewer that we should do more experiments that scale and generalize to different tasks in order to demonstrate the full potential of GGN, and we are still working hard on it.

---

### Decision · Program_Chairs · 2019-12-19

**Decision:**

Reject

**Comment:**

The article considers Gauss-Newton as a scalable second order alternative to train neural networks, and gives theoretical convergence rates and some experiments. The second order convergence results rely on the NTK and very wide networks. The reviewers pointed out that the method is of course not new, and suggested that comparison not only with SGD but also with methods such as Adam, natural gradients, KFAC, would be important, as well as additional experiments with other types of losses for classification problems and multidimensional outputs. The revision added preliminary experiments comparing with Adam and KFAC. Overall, I think that the article makes an interesting and relevant case that Gauss-Newton can be a competitive alternative for parameter optimization in neural networks. However, the experimental section could still be improved significantly. Therefore, I am recommending that the paper is not accepted at this time but revised to include more extensive experiments.